# End-to-end reconstruction meets data-driven regularization for inverse problems

**Subhadip Mukherjee**[*1], **Marcello Carioni**[*1], **Ozan Öktem**[2], and **Carola-Bibiane Schönlieb**[1]

[1]Department of Applied Mathematics and Theoretical Physics, University of Cambridge, UK
[2]Department of Mathematics, KTH–Royal institute of Technology, Sweden
[*]Equal contribution authors
Emails: {sm2467, mc2250, cbs31}@cam.ac.uk, ozan@kth.se

## Abstract

We propose a new approach for learning end-to-end reconstruction operators based on unpaired training data for ill-posed inverse problems. The proposed method combines the classical variational framework with iterative unrolling and essentially seeks to minimize a weighted combination of the expected distortion in the measurement space and the Wasserstein-1 distance between the distributions of the reconstruction and the ground-truth. More specifically, the regularizer in the variational setting is parametrized by a deep neural network and learned simultaneously with the unrolled reconstruction operator. The variational problem is then initialized with the output of the reconstruction network and solved iteratively till convergence. Notably, it takes significantly fewer iterations to converge as compared to variational methods, thanks to the excellent initialization obtained via the unrolled operator. The resulting approach combines the computational efficiency of end-to-end unrolled reconstruction with the well-posedness and noise-stability guarantees of the variational setting. Moreover, we demonstrate with the example of image reconstruction in X-ray computed tomography (CT) that our approach outperforms state-of-the-art unsupervised methods and that it outperforms or is at least on par with state-of-the-art supervised data-driven reconstruction approaches.

## 1 Introduction

Inverse problems are ubiquitous in imaging applications, wherein one seeks to recover an unknown model parameter $\boldsymbol{x} \in \mathbb{X}$ from its incomplete and noisy measurement, given by

$$\boldsymbol{y}^{\delta} = \mathcal{A}(\boldsymbol{x}) + \boldsymbol{e} \in \mathbb{Y}.$$

Here, the forward operator $\mathcal{A} : \mathbb{X} \to \mathbb{Y}$ models the measurement process in the absence of noise, and $\boldsymbol{e}$, with $\|\boldsymbol{e}\|_2 \leq \delta$, denotes the measurement noise. For example, in computed tomography (CT), the forward operator computes line integrals of $\boldsymbol{x}$ over a predetermined set of lines in $\mathbb{R}^3$ and the goal is to reconstruct $\boldsymbol{x}$ from its projections along these lines. Without any further information about $\boldsymbol{x}$, inverse problems are typically ill-posed, meaning that there could be several possible reconstructions that are consistent with the measured data, even without any noise.

The variational framework circumvents ill-posedness by encoding prior knowledge about $\boldsymbol{x}$ via a regularization functional $\mathcal{R} : \mathbb{X} \to \mathbb{R}$. In the variational setting, one solves

$$\min_{\boldsymbol{x} \in \mathbb{X}} \mathcal{L}_{\mathbb{Y}}(\boldsymbol{y}^{\delta}, \mathcal{A}(\boldsymbol{x})) + \lambda \, \mathcal{R}(\boldsymbol{x}), \qquad (1)$$

where $\mathcal{L}_{\mathbb{Y}} : \mathbb{Y} \times \mathbb{Y} \to \mathbb{R}^+$ measures data-fidelity and $\mathcal{R}$ penalizes undesirable or unlikely solutions. The penalty $\lambda > 0$ balances the regularization strength with the fidelity of the reconstruction. The variational problem (1) is said to be well-posed if it has a unique solution varying continuously in $\boldsymbol{y}^{\delta}$.

The success of deep learning in recent years has led to a surge of data-driven approaches for solving inverse problems [5], especially in imaging applications. These methods come broadly in two flavors: (i) end-to-end trained models that aim to directly map the measurement or a model-based noisy reconstruction to the corresponding true parameter [11, 2] and (ii) learned regularization methods that seek to find a data-adaptive regularizer instead of handcrafting it [15, 17, 13]. Techniques in both categories have their relative advantages and demerits. Specifically, end-to-end approaches offer fast reconstruction of astounding quality, but lack in terms of theoretical guarantees and need supervised data (i.e., pairs of input and target images) for training. On the contrary, learned regularization methods inherit the provable well-posedness properties of the variational setting and can be trained on unpaired training data, however the reconstruction entails solving a high-dimensional optimization problem, which is often slow and computationally demanding.

Our work derives ideas from learned optimization and adversarial machine learning, and makes an attempt to combine the best features of both aforementioned paradigms. In particular, the proposed method offers the flexibility of training on unpaired samples, produces fast reconstructions comparable to end-to-end supervised methods in quality, while enjoying the well-posedness and stability guarantees of the learned regularization framework. We first provide an overview of the recent literature on data-driven solutions for inverse problems before detailing our contributions.

## 1.1 Related works

End-to-end learned methods for inverse problems either map the measurement directly to the clean image [40, 22], or learn to eliminate artifacts from a reconstruction produced by a model-based technique [11]. Such approaches are data-intensive and may generalize poorly when trained on limited data. Iterative unrolling [20, 38, 1, 19, 12], with its origin in the seminal work by Gregor and LeCun on data-driven sparse coding [10], employs reconstruction networks that are inspired by optimization-based approaches and hence are interpretable. The unrolling paradigm enables one to encode the knowledge about the acquisition physics into the network architecture [2], thereby achieving data-efficiency. Nevertheless, end-to-end trained methods are supervised, and it is often challenging to obtain a large ensemble of paired data, especially in medical imaging applications.

Data-driven regularization methods aim to learn a regularizer in the variational setting instead of handcrafting it. Some notable approaches in this paradigm include adversarial regularization (AR) [17] and its convex counterpart [21], network Tikhonov (NETT) [15], total deep variation (TDV) [13], etc., wherein one explicitly parametrizes the regularization functional using a neural network. The regularization by denoising (RED) and more general Plug-and-play (PnP) approaches aim to solve inverse problems by using a pre-trained denoiser inside an algorithm for minimizing the variational objective [24, 25, 39, 36, 26, 16, 7, 31]. Such methods achieve remarkable performances for various inverse problems when equipped with sophisticated pre-trained denoisers [39, 36, 26, 16, 7], and are essentially equivalent to data-driven regularization, subject to additional constraints on the denoiser [23]. The deep image prior technique [33] is training-free, but it regularizes the solution by restricting it to be in the range of a generator and can thus be interpreted as a deep learning-based regularization scheme. It is relatively easier to analyze learned regularization schemes using classical functional analysis [28], but they tend to perform worse than end-to-end supervised methods. Moreover, these methods require solving a high-dimensional, potentially non-convex, variational optimization, leading to slow reconstruction with no provably convergent optimization algorithm in general.

Data-driven methods for inverse problems can be classified into different categories based on the training protocol. Supervised approaches need data in the form of pairs of noisy measurements (or some model-based reconstructions) and target images (or ground-truths) [2, 40, 11], while fully unsupervised methods require access to only the noisy measurements for training [29, 32, 8, 37]. Our approach lies in between these two extremes, as it requires samples from the marginal distributions of the measurements and the ground-truth images, but not from their joint distribution. Recently, such unpaired training approaches have been developed in [30, 14] based on the optimal transport theory. The training objective in [14] only seeks to match the ground-truth distribution via a Wasserstein generative adversarial network (GAN) without any data-consistency, whereas [30] shows that the cycle-GAN loss between the image and the measurement spaces is an upper bound on the optimal transport loss with a penalized least squares objective as the transportation cost. Unlike [30] and [14], our approach minimizes a fidelity term in the data space alone and not in the image space (together with an optimal transport-based loss), and allows for a more straightforward theoretical analysis using the standard variational framework.

## 1.2 Specific contributions

Our work seeks to combine iterative unrolling with data-adaptive regularization via an adversarial learning framework, and hence is referred to as unrolled adversarial regularization (UAR). The proposed approach learns a data-adaptive regularizer parametrized by a deep neural network, along with an iteratively unrolled reconstruction network that minimizes the corresponding expected variational loss in an adversarial setting. Unlike AR [17] where the undesirable images are taken as the pseudo-inverse reconstruction and kept fixed throughout the training, we update them with the output of the unrolled reconstruction network in each training step, and, in turn, use them to further improve the regularizer. Thanks to the Kantorovich-Rubinstein (KR) duality [4], the alternating learning strategy of the reconstruction and the regularizer networks is equivalent to minimizing the expected data-fidelity over the distribution of the measurement, penalized by the Wasserstein-1 distance between the distribution of the reconstruction and the ground-truth. Once trained, the iteratively unrolled operator produces a fast end-to-end reconstruction. We show that this efficient reconstruction can be improved further by a refinement step that involves minimizing the variational loss with the corresponding regularizer, starting from this initial estimate. The refinement step not only produces reconstructions that are superior to state-of-the-art unsupervised methods and are competitive with supervised methods, but also facilitates well-posedness and stability analyses akin to classical variational approaches [28]. Our theoretical results on the learned unrolled operator and the regularizer are corroborated by strong experimental evidence for the CT inverse problem, and the illustrative inpainting and denoising examples (see Sec. B in the supplementary document). The unpaired training framework for UAR is particularly useful for medical imaging applications, where it is in general challenging to collect a large corpus of paired noisy input and reference images.

## 2 The proposed unrolled adversarial regularization (UAR) approach

In this section, we give a brief mathematical background on optimal transport, followed by a detailed description of the UAR framework, including the training protocol and the network architectures.

### 2.1 Background on Optimal transport

Optimal transport theory [9, 34] has recently gained prominence in the context of measuring the distance between two probability measures. In particular, given two probability measures $\pi_1$ and $\pi_2$ on $\mathbb{R}^n$, the Wasserstein-1 distance between them is defined as

$$\mathbb{W}_1(\pi_1, \pi_2) := \inf_{\mu \in \Pi(\pi_1, \pi_2)} \int \|\boldsymbol{x}_1 - \boldsymbol{x}_2\|_2 \, \mathrm{d}\mu(\boldsymbol{x}_1, \boldsymbol{x}_2), \tag{2}$$

where $\Pi(\pi_1, \pi_2)$ denotes all transport plans having $\pi_1$ and $\pi_2$ as marginals. The Wasserstein distance has proven to be suitable for deep learning tasks, when the data is assumed to be concentrated on low-dimensional manifolds in $\mathbb{R}^n$. It has been shown that in such cases, the Wasserstein distance provides a usable gradient during training [4], as opposed to other popular divergence metrics.

By the KR duality, the Wasserstein-1 distance can be computed equivalently by solving a maximization problem over the space of 1-Lipschitz functions (denoted by $\mathbb{L}_1$) as

$$\mathbb{W}_1(\pi_1, \pi_2) = \sup_{\mathcal{R} \in \mathbb{L}_1} \int \mathcal{R}(\boldsymbol{x}_1) \, \mathrm{d}\pi_1(\boldsymbol{x}_1) - \int \mathcal{R}(\boldsymbol{x}_2) \, \mathrm{d}\pi_2(\boldsymbol{x}_2), \tag{3}$$

provided that $\pi_1$ and $\pi_2$ have compact support [27]. Finally, we recall the definition of push-forward of probability measures, which is used extensively in our theoretical exposition. Given a probability measure $\pi$ on $\mathscr{A}$ and a map $T : \mathscr{A} \to \mathscr{B}$, the push-forward of $\pi$ by $T$ (denoted as $T_\#\pi$) is defined as a probability measure on $\mathscr{B}$ such that $T_\#\pi(B) = \pi(T^{-1}(B))$, for all measurable $B \subset \mathscr{B}$.

### 2.2 Training strategy and network parametrization for UAR

The principal idea behind UAR is to learn an unrolled deep network $\mathcal{G}_\phi : \mathbb{Y} \to \mathbb{X}$ for reconstruction, together with a regularization functional $\mathcal{R}_\theta : \mathbb{X} \to \mathbb{R}$ parametrized by another convolutional neural network (CNN). The role of $\mathcal{R}_\theta$ is to discern ground-truth images from images produced by $\mathcal{G}_\phi$, while $\mathcal{G}_\phi$ learns to minimize the variational loss with $\mathcal{R}_\theta$ as the regularizer. As the images produced

---

**Algorithm 1** Learning unrolled adversarial regularization (UAR).

---

**1. Input:** Training data-set $\{\boldsymbol{x}_i\}_{i=1}^N \sim \pi_x$ and $\{\boldsymbol{y}_j\}_{j=1}^N \sim \pi_{y^\delta}$, initial reconstruction network parameter $\phi$ and regularizer parameter $\theta$, batch-size $n_b = 1$, penalty $\lambda = 0.1$, gradient penalty $\lambda_{\text{gp}} = 10.0$, Adam optimizer parameters $(\beta_1, \beta_2) = (0.50, 0.99)$.

**2. Learn a baseline regularizer**:

- **for training steps** $k = 1, 2, \cdots, 10\,\frac{N}{n_b}$, **do:** (i.e., # epochs = 10, # mini-batches = $\frac{N}{n_b}$)

  - Sample $\boldsymbol{x}_j \sim \pi_x$, $\boldsymbol{y}_j \sim \pi_{y^\delta}$, and $\epsilon_j \sim$ uniform $[0,1]$; for $1 \leq j \leq n_b$. Compute $\boldsymbol{u}_j = \mathcal{A}^\dagger(\boldsymbol{y}_j)$ and $\boldsymbol{x}_j^{(\epsilon)} = \epsilon_j \boldsymbol{x}_j + (1 - \epsilon_j)\,\boldsymbol{u}_j$.

  - $\theta \leftarrow \text{Adam}_{\eta, \beta_1, \beta_2}(\theta, \nabla \tilde{J}_1(\theta))$, where $\eta = 10^{-4}$, and

$$\tilde{J}_1(\theta) = \frac{1}{n_b} \sum_{j=1}^{n_b} \left[ \mathcal{R}_\theta\left(\boldsymbol{x}_j\right) - \mathcal{R}_\theta\left(\boldsymbol{u}_j\right) + \lambda_{\text{gp}} \left( \left\| \nabla \mathcal{R}_\theta\left(\boldsymbol{x}_j^{(\epsilon)}\right) \right\|_2 - 1 \right)^2 \right].$$

**3. Learn a baseline reconstruction operator**: (with 5 epochs, $\frac{N}{n_b}$ batches per epoch)

- **for training steps** $k = 1, 2, \cdots, 5\,\frac{N}{n_b}$, **do:**

  - Sample $\boldsymbol{y}_j \sim \pi_{y^\delta}$, and compute $\tilde{J}_2(\phi) = \frac{1}{n_b} \sum_{j=1}^{n_b} \left\| \boldsymbol{y}_j - \mathcal{A}\left(\mathcal{G}_\phi(\boldsymbol{y}_j)\right) \right\|_2^2 + \lambda\, \mathcal{R}_\theta\left(\mathcal{G}_\phi(\boldsymbol{y}_j)\right)$.

  - $\phi \leftarrow \text{Adam}_{\eta, \beta_1, \beta_2}(\phi, \nabla \tilde{J}_2(\phi))$, with step-size $\eta = 10^{-4}$.

**4. Jointly train $\mathcal{R}_\theta$ and $\mathcal{G}_\phi$ adversarially**: (over 25 epochs, with $\frac{N}{n_b}$ batches in each epoch)

- **for** $k = 1, 2, \cdots, 25\,\frac{N}{n_b}$, **do:**

  - Sample $\boldsymbol{x}_j$, $\boldsymbol{y}_j$, and $\epsilon_j \sim$ uniform $[0,1]$; for $1 \leq j \leq n_b$. Compute $\boldsymbol{u}_j = \mathcal{G}_\phi(\boldsymbol{y}_j)$ and $\boldsymbol{x}_j^{(\epsilon)} = \epsilon_j \boldsymbol{x}_j + (1 - \epsilon_j)\,\boldsymbol{u}_j$.

  - $\theta \leftarrow \text{Adam}_{\eta, \beta_1, \beta_2}(\theta, \nabla \tilde{J}_1(\theta))$, where $\tilde{J}_1(\theta)$ is as in Step 2, with $\eta = 2 \times 10^{-5}$.

  - Update $\phi \leftarrow \text{Adam}_{\eta, \beta_1, \beta_2}(\phi, \nabla \tilde{J}_2(\phi))$ twice, with $\tilde{J}_2(\phi)$ as in Step 3, and $\eta = 2 \times 10^{-5}$.

**5. Output:** The trained networks $\mathcal{G}_\phi$ and $\mathcal{R}_\theta$.

---

by $\mathcal{G}_\phi$ get better, $\mathcal{R}_\theta$ faces a progressively harder task of telling them apart from the ground-truth images, thus leading to an improved regularizer. On the other hand, as the regularizer improves, the quality of reconstructions obtained using $\mathcal{G}_\phi$ improves as a consequence. Thus, $\mathcal{G}_\phi$ and $\mathcal{R}_\theta$ help each other improve as the training progresses via an alternating update scheme.

### 2.2.1 Adversarial training

Let us denote by $\pi_x$ the ground-truth distribution and by $\pi_{y^\delta}$ the distribution of the noisy measurement. The UAR approach seeks to learn $\mathcal{G}_\phi$ and $\mathcal{R}_\theta$ simultaneously starting from an appropriate initialization. At the $k^{\text{th}}$ iteration of training, the parameters $\phi$ of the reconstruction network are updated as

$$\phi_k \in \arg\min_\phi J_k^{(1)}(\phi), \text{ where } J_k^{(1)}(\phi) := \mathbb{E}_{\pi_{y^\delta}} \left[ \left\| \mathcal{A}(\mathcal{G}_\phi(\boldsymbol{y}^\delta)) - \boldsymbol{y}^\delta \right\|_2^2 + \lambda\, \mathcal{R}_{\theta_k}(\mathcal{G}_\phi(\boldsymbol{y}^\delta)) \right], \quad (4)$$

for a fixed regularizer parameter $\theta_k$. Subsequently, the regularizer parameters are updated as

$$\theta_{k+1} \in \arg\max_{\theta : \mathcal{R}_\theta \in \mathbb{L}_1} J_k^{(2)}(\theta), \text{ where } J_k^{(2)}(\theta) := \mathbb{E}_{\pi_{y^\delta}} \left[ \mathcal{R}_\theta\left(\mathcal{G}_{\phi_k}(\boldsymbol{y}^\delta)\right) \right] - \mathbb{E}_{\pi_x} \left[ \mathcal{R}_\theta\left(\boldsymbol{x}\right) \right]. \quad (5)$$

The learning protocol for UAR only requires unpaired samples of $\boldsymbol{x}$ and $\boldsymbol{y}^\delta$, since the loss functionals $J_k^{(1)}(\phi)$ and $J_k^{(2)}(\theta)$ can be computed based solely on the marginals $\pi_x$ and $\pi_{y^\delta}$. The alternating update algorithm in (4) and (5) essentially seeks to solve the min-max variational problem given by

$$\min_\phi \max_{\theta : \mathcal{R}_\theta \in \mathbb{L}_1} \mathbb{E}_{\pi_{y^\delta}} \left\| \mathcal{A}(\mathcal{G}_\phi(\boldsymbol{y}^\delta)) - \boldsymbol{y}^\delta \right\|_2^2 + \lambda \left( \mathbb{E}_{\pi_{y^\delta}} \left[ \mathcal{R}_\theta\left(\mathcal{G}_\phi(\boldsymbol{y}^\delta)\right) \right] - \mathbb{E}_{\pi_x} \left[ \mathcal{R}_\theta\left(\boldsymbol{x}\right) \right] \right). \quad (6)$$

Thanks to KR duality in (3) and the definition of push-forward, (6) can be reformulated as

$$\min_\phi \mathbb{E}_{\pi_{y^\delta}} \left\| \mathcal{A}(\mathcal{G}_\phi(\boldsymbol{y}^\delta)) - \boldsymbol{y}^\delta \right\|_2^2 + \lambda \mathbb{W}_1 \left( (\mathcal{G}_\phi)_\# \pi_{y^\delta}, \pi_x \right). \quad (7)$$

We refer the reader to Section 3 for a mathematically rigorous statement of this equivalence as well as for a well-posedness theory of the problem in (7). Note that the equivalence of the alternating minimization procedure and the variational problem in (7) holds only if the regularizer is fully optimized in every iteration. Nevertheless, in practice, the reconstruction and regularizer networks are not fully optimized in every iteration. Instead, one refines the parameters by performing one (or a few) Adam updates on the corresponding loss functionals. Notably, if $\mathbb{W}_1\left((\mathcal{G}_{\phi_k})_{\#}\pi_{y^\delta}, \pi_x\right) = 0$, i.e., the parameters of $\mathcal{G}$ are such that the reconstructed images match the ground-truth in distribution, the loss functional $J_k^{(2)}(\theta)$ and its gradient vanish, leading to no further update of $\theta$. Thus, both networks stop updating when the outputs of $\mathcal{G}_\phi$ are indistinguishable from the ground-truth images. The concrete training steps are listed in Algorithm 1[1].

### 2.2.2 Variational regularization as a refinement step

The unrolled operator $\mathcal{G}_{\phi^*}$ trained by solving the min-max problem in (6) provides reasonably good reconstruction when evaluated on X-ray CT, and already outperforms state-of-the-art unsupervised methods (c.f. Section 4). We demonstrate that the regularizer $\mathcal{R}_{\theta^*}$ obtained together with $\mathcal{G}_{\phi^*}$ by solving (6) can be used in the variational framework to further improve the quality of the end-to-end reconstruction $\mathcal{G}_{\phi^*}(\boldsymbol{y}^\delta)$ for a given $\boldsymbol{y}^\delta \in \mathbb{Y}$. Specifically, we solve the variational problem

$$\min_{\boldsymbol{x} \in \mathbb{X}} \| \mathcal{A}(\boldsymbol{x}) - \boldsymbol{y}^\delta \|_2 + \lambda' \left(\mathcal{R}_{\theta^*}(\boldsymbol{x}) + \sigma\|\boldsymbol{x}\|_2^2\right), \tag{8}$$

where $\lambda', \sigma \geq 0$, by applying gradient descent, initialized with $\mathcal{G}_{\phi^*}(\boldsymbol{y}^\delta)$. The additional Tikhonov term in (8) ensures coercivity of the overall regularizer, making it amenable to the standard well-posedness analysis [28]. Practically, it improves the stability of the gradient descent optimizer for (8). Nevertheless, one essentially gets the same reconstruction with $\sigma = 0$ subject to early stopping (100 iterations). Notably, the fidelity term in (8) is the $\ell_2$ distance, instead of the squared-$\ell_2$ fidelity. We have empirically observed that this choice of the fidelity term improves the quality of the reconstruction, possibly due to the higher gradient of the objective in the initial solution $\mathcal{G}_{\phi^*}(\boldsymbol{y}^\delta)$. Since the end-to-end reconstruction gives an excellent initial point, it takes significantly fewer iterations for gradient-descent to recover the optimal solution to (8), and therefore UAR retains its edge in reconstruction time over fully variational approaches with learned regularizers (e.g., AR [17] or its convex version [21]).

### 2.2.3 Architectures of the iteratively unrolled generator and the deep regularizer

The objective of $\mathcal{G}_\phi$ is to approximate the minimizer of the variational loss with $\mathcal{R}_\theta$ as the regularizer. Therefore, an iterative unrolling strategy akin to [2] is adopted for parametrizing $\mathcal{G}_\phi$. Iterative unrolling aims to approximate the variational minimizer via a primal-dual-style algorithm [6], with the proximal operators in the image and measurement spaces replaced with trainable CNNs. Although the variational loss in our case is non-convex, this parametrization for $\mathcal{G}_\phi$ is chosen because of its expressive power over a generic network. Initialized with $\boldsymbol{x}^{(0)} = \mathcal{A}^\dagger \boldsymbol{y}^\delta$ and $\boldsymbol{h}^{(0)} = \boldsymbol{0}$, $\mathcal{G}_\phi$ produces a reconstruction $\boldsymbol{x}^{(L)}$ by iteratively applying the CNNs $\Lambda_{\phi_{\mathrm{p}}^{(\ell)}}$ and $\Gamma_{\phi_{\mathrm{d}}^{(\ell)}}$ in $\mathbb{X}$ and $\mathbb{Y}$, respectively:

$$\boldsymbol{h}^{(\ell+1)} = \Gamma_{\phi_{\mathrm{d}}^{(\ell)}}\left(\boldsymbol{h}^{(\ell)}, \sigma^{(\ell)}\mathcal{A}(\boldsymbol{x}^\ell), \boldsymbol{y}^\delta\right), \text{ and } \boldsymbol{x}^{(\ell+1)} = \Lambda_{\phi_{\mathrm{p}}^{(\ell)}}\left(\boldsymbol{x}^{(\ell)}, \tau^{(\ell)}\mathcal{A}^*(\boldsymbol{h}^{\ell+1})\right), 0 \leq \ell \leq L-1.$$

The step-size parameters $\sigma^{(\ell)}$ and $\tau^{(\ell)}$ are also made learnable and initialized as $\sigma^{(\ell)} = \tau^{(\ell)} = 0.01$ for each layer $\ell$. All learnable parameters in the generator are denoted by the shorthand notation $\phi$ for brevity. The number of layers $L$ is typically much smaller (we take $L = 20$) than the number of iterations needed by an iterative primal-dual scheme to converge, thus expediting the reconstruction by two orders of magnitude (as compared to variational methods) once trained.

The regularizer $\mathcal{R}_\theta$ is taken as a deep CNN with six convolutional layers, followed by one average-pooling and two dense layers in the end.

## 3 Theoretical results

The theoretical properties of UAR are stated in this section and their proofs are provided in the supplementary document. Throughout this section, we assume that $\mathbb{X} = \mathbb{R}^n$ and $\mathbb{Y} = \mathbb{R}^k$, and

---

[1]Codes at `https://github.com/Subhadip-1/unrolling_meets_data_driven_regularization`.

A1. $\pi_x$ is compactly supported and $\pi_{y^\delta}$ is supported on a compact set $\mathcal{K} \subset \mathbb{R}^k$ for every $\delta \geq 0$.

We then consider the following problem:

$$\inf_{\phi} \sup_{\mathcal{R} \in \mathbb{L}_1} J_1\left(\mathcal{G}_\phi, \mathcal{R} | \lambda, \pi_{y^\delta}\right) := \mathbb{E}_{\pi_{y^\delta}} \left\| \boldsymbol{y}^\delta - \mathcal{A}\,\mathcal{G}_\phi(\boldsymbol{y}^\delta) \right\|_2^2 + \lambda \left( \mathbb{E}_{\pi_{y^\delta}} \left[ \mathcal{R}(\mathcal{G}_\phi(\boldsymbol{y}^\delta)) \right] - \mathbb{E}_{\pi_x} \left[ \mathcal{R}(\boldsymbol{x}) \right] \right).$$

$$(9)$$

Problem (9) is identical to the min-max variational problem defined in (6), with the only difference that the maximization in $\mathcal{R}$ is performed over the space of all 1-Lipschitz functions. Basically, we consider the theoretical limiting case where the neural networks $\mathcal{R}_\theta$ are expressive enough to approximate all functions in $\mathbb{L}_1$ with arbitrary accuracy. We make the following assumptions on $\mathcal{G}_\phi$:

A2. $\mathcal{G}_\phi$ is parametrized over a finite dimensional compact set $K$, i.e. $\phi \in K$.

A3. $\mathcal{G}_{\phi_n} \to \mathcal{G}_\phi$ pointwise whenever $\phi_n \to \phi$.

A4. $\sup_{\phi \in K} \| \mathcal{G}_\phi \|_\infty < \infty$.

Assumptions A2-A4 are satisfied, for instance, when $\mathcal{G}_\phi$ is parametrized by a neural network whose weights are kept bounded during training. These assumptions apply to all results in this section.

## 3.1 Well-posedness of the adversarial loss and noise stability

Here, we prove well-posedness and stability to noise for the optimal reconstructions. As a consequence of the KR duality, (9) can be equivalently expressed as

$$\inf_{\phi} J_2\left(\mathcal{G}_\phi | \lambda, \pi_{y^\delta}\right) := \mathbb{E}_{\pi_{y^\delta}} \left\| \boldsymbol{y}^\delta - \mathcal{A}\,\mathcal{G}_\phi(\boldsymbol{y}^\delta) \right\|_2^2 + \lambda \mathbb{W}_1(\pi_x, (\mathcal{G}_\phi)_\# \pi_{y^\delta}). \tag{10}$$

In the next theorem, we state this equivalence, showing the existence of an optimal $\mathcal{G}_\phi$ and $\mathcal{R}$ for (9).

**Theorem 1.** *Problems* (9) *and* (10) *admit an optimal solution and*

$$\inf_{\phi} \sup_{\mathcal{R} \in \mathbb{L}_1} J_1\left(\mathcal{G}_\phi, \mathcal{R} | \lambda, \pi_{y^\delta}\right) = \inf_{\phi} J_2\left(\mathcal{G}_\phi | \lambda, \pi_{y^\delta}\right). \tag{11}$$

*Moreover, if* $(\mathcal{G}_{\phi^*}, \mathcal{R}^*)$ *is optimal for* (9), *then* $\mathcal{G}_{\phi^*}$ *is optimal for* (10). *Conversely, if* $\mathcal{G}_{\phi^*}$ *is optimal for* (10), *then* $(\mathcal{G}_{\phi^*}, \mathcal{R}^*)$ *is optimal for* (9), *for all* $\mathcal{R}^* \in \arg\max_{\mathcal{R} \in \mathbb{L}_1} \mathbb{E}_{\pi_{y^\delta}} \left[ \mathcal{R}(\mathcal{G}_{\phi^*}(\boldsymbol{y}^\delta)) \right] - \mathbb{E}_{\pi_x} \left[ \mathcal{R}(\boldsymbol{x}) \right]$.

Next, we study the stability of the optimal reconstruction $\mathcal{G}_{\phi^*}$ to noise. We consider $\mathcal{G}_{\phi_n}$, where

$$\phi_n \in \arg\inf_{\phi} J_2\left(\mathcal{G}_\phi | \lambda, \pi_{y^{\delta_n}}\right), \tag{12}$$

and show that $\mathcal{G}_{\phi_n} \to \mathcal{G}_{\phi^*}$ as $\delta_n \to \delta$, thus establishing noise-stability of the unrolled reconstruction.

**Theorem 2** (Stability to noise). *Suppose, for given a sequence of noise levels* $\delta_n \to \delta \in [0, \infty)$, *it holds that* $\pi_{y^{\delta_n}} \to \pi_{y^\delta}$ *in total variation. Then, with* $\phi_n$ *as in* (12), $\mathcal{G}_{\phi_n} \to \mathcal{G}_{\phi^*}$ *up to sub-sequences.*

## 3.2 Effect of $\lambda$ on the end-to-end reconstruction

In order to analyze the effect of the parameter $\lambda$ in (10) on the resulting reconstruction $\mathcal{G}_{\phi^*}$, it is convenient to introduce the following two sets:

$$\Phi_{\mathcal{L}} := \left\{ \phi : \mathbb{E}_{\pi_{y^\delta}} \left\| \boldsymbol{y}^\delta - \mathcal{A}\,\mathcal{G}_\phi(\boldsymbol{y}^\delta) \right\|_2^2 = 0 \right\} \text{ and } \Phi_{\mathbb{W}} := \left\{ \phi : (\mathcal{G}_\phi)_\# \pi_{y^\delta} = \pi_x \right\}.$$

We assume that both $\Phi_{\mathcal{L}}$ and $\Phi_{\mathbb{W}}$ are non-empty, which is tantamount to asking that the parametrization of the end-to-end reconstruction operator is expressive enough to approximate a right inverse of $\mathcal{A}$ ($\Phi_{\mathcal{L}} \neq \emptyset$) and a transport map from $\pi_{y^\delta}$ to $\pi_x$ ($\Phi_{\mathbb{W}} \neq \emptyset$), and therefore is not very restrictive (keeping in view the excellent approximation power of unrolled deep architectures).

**Proposition 1.** *Let* $\mathcal{G}_{\phi^*}$ *be a minimizer for* (10). *Then, it holds that*

- $\mathbb{E}_{\pi_y} \left\| \boldsymbol{y}^\delta - \mathcal{A}\,\mathcal{G}_{\phi^*}(\boldsymbol{y}^\delta) \right\|_2^2 \leq \lambda \mathbb{W}_1(\pi_x, (\mathcal{G}_\phi)_\# \pi_{y^\delta}),$ *for every* $\phi \in \Phi_{\mathcal{L}}$.

- $\mathbb{W}_1(\pi_x, (\mathcal{G}_{\phi^*})_\# \pi_{y^\delta}) \leq \dfrac{1}{\lambda} \mathbb{E}_{\pi_{y^\delta}} \left\| \boldsymbol{y}^\delta - \mathcal{A}\,\mathcal{G}_\phi(\boldsymbol{y}^\delta) \right\|_2^2,$ *for every* $\phi \in \Phi_{\mathbb{W}}$.

| method | | PSNR (dB) | SSIM | # param. | reconstruction time (ms) |
|---|---|---|---|---|---|
| *Classical model-based methods* | | | | | |
| FBP | | $21.28 \pm 0.13$ | $0.20 \pm 0.02$ | 1 | $37.0 \pm 4.6$ |
| TV | | $30.31 \pm 0.52$ | $0.78 \pm 0.01$ | 1 | $28371.4 \pm 1281.5$ |
| *Trained on paired data* | | | | | |
| U-Net | | $34.50 \pm 0.65$ | $0.90 \pm 0.01$ | 7215233 | $44.4 \pm 12.5$ |
| LPD | | $35.69 \pm 0.60$ | $0.91 \pm 0.01$ | 1138720 | $279.8 \pm 12.8$ |
| *Trained on unpaired data* | | | | | |
| AR | | $33.84 \pm 0.63$ | $0.86 \pm 0.01$ | 19338465 | $22567.1 \pm 309.7$ |
| ACR | | $31.55 \pm 0.54$ | $0.85 \pm 0.01$ | 606610 | $109952.4 \pm 497.8$ |
| UAR | $\lambda = 0.001$ | $21.59 \pm 0.11$ | $0.22 \pm 0.02$ | 20477186 | $252.7 \pm 13.3$ |
| | $\lambda = 0.01$ | $25.25 \pm 0.08$ | $0.37 \pm 0.01$ | | |
| | $\lambda = 0.1$ | $34.35 \pm 0.66$ | $0.88 \pm 0.01$ | | |
| | $\lambda = 1.0$ | $33.27 \pm 0.76$ | $0.87 \pm 0.01$ | | |
| UAR with refinement | $\lambda = \lambda' = 0.1$ | $34.77 \pm 0.67$ | $0.90 \pm 0.01$ | – | $5863.3 \pm 106.1$ |

Table 1: Average PSNR and SSIM (with their standard deviations) for different reconstruction methods. The reconstruction times and the number of learnable parameters are also indicated. Without any refinement, UAR outperforms AR and ACR in reconstruction quality and reduces the reconstruction time by a couple of orders of magnitude. With refinement, UAR narrowly outperforms supervised U-net post-processing, and the reconstruction is roughly four times faster than AR.

The previous proposition shows in a quantitative way that for small $\lambda$, the optimal $\mathcal{G}_{\phi^*}$ has less expected distortion in the measurement space as the quantity $\mathbb{E}_{\pi_{y^\delta}} \left\| \boldsymbol{y}^\delta - \mathcal{A}\,\mathcal{G}_{\phi^*}(\boldsymbol{y}^\delta) \right\|_2^2$ is small. On the other hand, if $\lambda$ is large, then the optimal $\mathcal{G}_{\phi^*}$ maps $\pi_{y^\delta}$ closer to the ground-truth distribution $\pi_x$; as the quantity $\mathbb{W}(\pi_x, (\mathcal{G}_{\phi^*})_{\#}\pi_{y^\delta})$ is small. Therefore, the regularization is stronger in this case.

We extend this analysis by studying the behavior of the unrolled reconstruction as $\lambda$ converges to $0$ and to $+\infty$. Consider a sequence of parameters $\lambda_n > 0$ and the minimizer of the objective in (10) with parameter $\lambda_n$:

$$\phi'_n \in \arg\inf_{\phi} J_2\left(\mathcal{G}_\phi | \lambda_n, \pi_{y^\delta}\right). \tag{13}$$

**Theorem 3.** *Let $\lambda_n \to 0$. Then, there exists $\phi_1^* \in \arg\min_{\phi \in \Phi_{\mathcal{L}}} \mathbb{W}_1(\pi_x, (\mathcal{G}_\phi)_{\#}\pi_{y^\delta})$ such that $\mathcal{G}_{\phi'_n} \to \mathcal{G}_{\phi_1^*}$ up to sub-sequences, and $\lim_{n\to\infty} \frac{1}{\lambda_n}\inf_{\phi} J_2\left(\mathcal{G}_\phi | \lambda_n, \pi_{y^\delta}\right) = \mathbb{W}_1(\pi_x, (\mathcal{G}_{\phi_1^*})_{\#}\pi_{y^\delta})$.*

**Theorem 4.** *Let $\lambda_n \to +\infty$. Then, there exists $\phi_2^* \in \arg\min_{\phi \in \Phi_{\mathbb{W}}} \mathbb{E}_{\pi_{y^\delta}} \left\| \boldsymbol{y}^\delta - \mathcal{A}\,\mathcal{G}_\phi(\boldsymbol{y}^\delta) \right\|_2^2$ such that $\mathcal{G}_{\phi'_n} \to \mathcal{G}_{\phi_2^*}$ up to sub-sequences, and $\lim_{n\to\infty} \inf_{\phi} J_2\left(\mathcal{G}_\phi | \lambda_n, \pi_{y^\delta}\right) = \mathbb{E}_{\pi_{y^\delta}} \left\| \boldsymbol{y}^\delta - \mathcal{A}\,\mathcal{G}_{\phi_2^*}(\boldsymbol{y}^\delta) \right\|_2^2$.*

Theorems 3 and 4 characterize the optimal end-to-end reconstruction $\mathcal{G}_{\phi^*}$ as $\lambda \to 0$ and $\lambda \to \infty$, respectively. Specifically, if $\lambda \to 0$, $\mathcal{G}_{\phi^*}$ minimizes the Wasserstein distance between reconstruction and ground-truth among all the reconstruction operators that achieve zero expected data-distortion. In particular, $\mathcal{G}_{\phi^*}$ is close to the right inverse of $\mathcal{A}$ that minimizes the Wasserstein distance. Therefore, when $\lambda$ is very small, we expect to obtain a reconstruction that is close to the unregularized solution in quality. If $\lambda \to \infty$ on the other hand, the operator $\mathcal{G}_{\phi^*}$ is close to a transport map between $\pi_x$ and $\pi_{y^\delta}$, i.e., $(\mathcal{G}_{\phi^*})_{\#}\pi_{y^\delta} = \pi_x$, which minimizes the expected data-distortion. Therefore the reconstruction produces realistic images, but they are not consistent with the measurement. These theoretical observations are corroborated by the numerical results (c.f. Section 4, Fig. 2). One has to thus select a $\lambda$ that optimally trades-off data-distortion with the Wasserstein distance to achieve the best reconstruction performance.

## 3.3 End-to-end reconstruction vis-à-vis the variational solution

The goal of this section is two-fold. Firstly, we theoretically justify the fact that the end-to-end reconstruction performs well, despite minimizing the expected loss over the distribution $\pi_{y^\delta}$. Secondly, we analyze the role of the regularizer in the variational setting in refining the end-to-end reconstruction.

It is important to remark that the the end-to-end reconstruction is trained with respect to the average variational loss computed using samples from $\pi_{y^\delta}$ and $\pi_x$. Therefore, the end-to-end reconstruction cannot learn a point-wise correspondence between measurement and model parameter, but only a distributional correspondence. Despite that, the end-to-end reconstruction achieves excellent performance for a given measurement $y^\delta$. A justification of this phenomenon is given formally by the next proposition.

**Proposition 2.** *Let $(\mathcal{G}_{\phi^*}, \mathcal{R}^*)$ be an optimal pair for* (9) *such that $\mathcal{R}^* \geq 0$ almost everywhere under* $(\mathcal{G}_\phi)_\# \, \pi_{y^\delta}$. *Define $M_1 := \mathbb{E}_{\pi_{y^\delta}} \left\| y^\delta - \mathcal{A}\,\mathcal{G}_{\phi^*}(y^\delta) \right\|_2^2$ and $M_2 := \mathbb{W}_1(\pi_x, (\mathcal{G}_{\phi^*})_\# \pi_{y^\delta})$. Then, the following two upper bounds hold for every $\eta > 0$:*

- $\mathbb{P}_{\pi_{y^\delta}} \left\{ y^\delta : \left\| y^\delta - \mathcal{A}\,\mathcal{G}_{\phi^*}(y^\delta) \right\|_2^2 \geq \eta \right\} \leq \frac{M_1}{\eta}$.

- *Suppose, $\mathcal{R}^*(x) = 0$ for $\pi_x$-almost every $x$. Then, $\mathbb{P}_{(\mathcal{G}_{\phi^*})_\# \pi_{y^\delta}} \left\{ x : \mathcal{R}^*(x) \geq \eta \right\} \leq \frac{M_2}{\eta}$.*

Proposition 2 provides an estimate in probability of the sets $\{ y^\delta : \left\| y^\delta - \mathcal{A}\,\mathcal{G}_{\phi^*}(y^\delta) \right\|_2^2 \geq \eta \}$ and $\{ x : \mathcal{R}^*(x) \geq \eta \}$. In particular, if $M_1$ is small, then $\left\| y^\delta - \mathcal{A}\,\mathcal{G}_{\phi^*}(y^\delta) \right\|_2^2$ is small in probability. If instead $M_2$ is small, then $\mathcal{R}^*(x)$ is small in probability on the support of $(\mathcal{G}_{\phi^*})_\# \pi_{y^\delta}$, implying that samples $\mathcal{G}_{\phi^*}(y^\delta)$ are difficult to distinguish from the ground-truth. We remark that the assumption $\mathcal{R}^*(x) = 0$ can be justified using a data manifold assumption as in Section 3.3. of [17]. We now analyze the role of the regularizer $\mathcal{R}^*$ in the optimization of the variational problem (8) that refines the end-to-end reconstruction $\mathcal{G}_{\phi^*}$. We rely on a similar distributional analysis as the one performed in [17]. For $\eta > 0$, consider the transformation by a gradient-descent step on $\mathcal{R}^*$ given by $g_\eta(x) = x - \eta \nabla \mathcal{R}^*(x)$. Using the shorthand $\pi_{\mathcal{G}^*} := (\mathcal{G}_{\phi^*})_\# \pi_{y^\delta}$, and by denoting the distribution of $g_\eta(x)$ as $\pi_\eta := (g_\eta)_\# \pi_{\mathcal{G}^*}$ for $x \sim \pi_{\mathcal{G}^*}$, we have the following theorem.

**Theorem 5** ([17]). *Suppose that $\eta \to \mathbb{W}_1(\pi_\eta, \pi_x)$ is differentiable at $\eta = 0$. Then, the derivative at $\eta = 0$ satisfies $\frac{\mathrm{d}}{\mathrm{d}\eta} \mathbb{W}_1(\pi_\eta, \pi_x) \Big|_{\eta=0} = -\mathbb{E}_{\pi_{\mathcal{G}^*}} \| \nabla \mathcal{R}^*(x) \|_2^2$.*

This theorem states that a gradient-descent step performed on $\mathcal{R}^*$ at $x = \mathcal{G}_{\phi^*}(y^\delta)$ decreases the Wasserstein distance with respect to the ground-truth distribution $\pi_x$. Therefore, if the gradient-descent step to solve the variational problem (8) is initialized with the reconstruction $\mathcal{G}_{\phi^*}(y^\delta)$, the next iterate gets pushed closer to the ground-truth distribution $\pi_x$. We stress that this property holds because of the chosen initialization point, due to the relation between $\mathcal{R}^*$ and $\mathcal{G}_{\phi^*}$. For a different initialization, this property may not hold.

## 4 Numerical results

On the application front, we consider the prototypical inverse problem of CT reconstruction from noisy sparse-view projections. The abdominal CT scans for 10 patients, made publicly available by the Mayo-Clinic for the low-dose CT grand challenge [18], were used in our numerical experiments. Specifically, 2250 2D slices of size $512 \times 512$ corresponding to 9 patients were used to train the models, while 128 slices from the remaining one patient were used for evaluation. The projections were simulated in ODL [3] using a parallel-beam geometry with 200 uniformly spaced angular positions of the source, with 400 lines per angle. Subsequently, Gaussian noise with standard deviation $\sigma_e = 2.0$ was added to the projection data to simulate noisy sinograms. Notably, the CT projection data are typically corrupted by Poisson noise, but the data-likelihood for Poisson noise is not the $\ell_2^2$ distance and is therefore not amenable to the theoretical analysis presented in Sec. 3. Hence, we simulate the noisy projection data by adding Gaussian noise to demonstrate a proof-of-concept for the UAR method. Extending the theoretical results to deal with Poisson noise in the data remains to be undertaken as a future work. Additionally, since the UAR approach seeks to

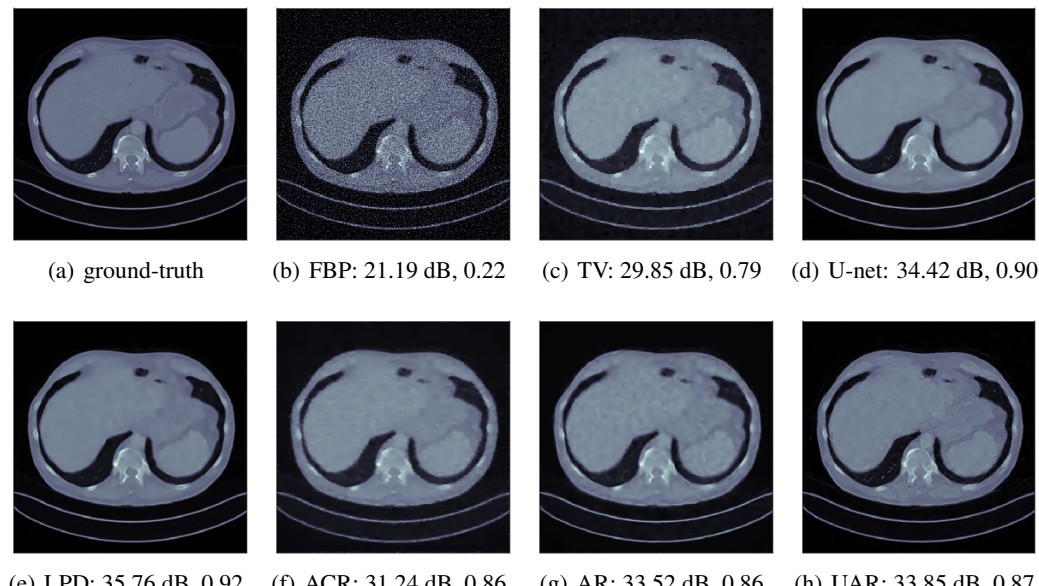

(a) ground-truth    (b) FBP: 21.19 dB, 0.22    (c) TV: 29.85 dB, 0.79    (d) U-net: 34.42 dB, 0.90

(e) LPD: 35.76 dB, 0.92    (f) ACR: 31.24 dB, 0.86    (g) AR: 33.52 dB, 0.86    (h) UAR: 33.85 dB, 0.87

Figure 1: Reconstruction on Mayo clinic data. UAR achieves better reconstruction quality than AR and ACR, while significantly reducing the reconstruction time (c.f. Table 1). The reduction in reconstruction time comes at the expense of higher training complexity as compared to AR. The numbers below the images indicate the corresponding PSNR and SSIM scores.



(a) ground-truth    (b) 21.60, 0.21    (c) 25.33, 0.37    (d) 34.65, 0.88    (e) 33.96, 0.88

Figure 2: UAR reconstruction for different $\lambda$. The values of $\lambda$ for (b), (c), (d), and (e) are 0.001, 0.01, 0.1, and 1.0, respectively. For $\lambda \to 0$, the unrolled generator seeks to find the minimizer of the expected data-fidelity loss, hence the reconstruction looks similar to FBP. The corresponding PSNR (dB) and SSIM with respect to the ground-truth are indicated below the images.

minimize a variational loss, which includes a data-fidelity term arising from the noise distribution, one needs to systematically investigate the robustness of UAR to the change in noise statistics to make it fully applicable to CT reconstruction from real low-dose projection data.

The proposed UAR method is compared with two classical model-based approaches for CT, namely filtered back-projection (FBP) and total variation (TV). The LPD method [2] and U-net-based post-processing [11] of FBP are chosen as two supervised approaches for comparison. The AR approach [17] and its convex variant [21], referred to as adversarial convex regularizer (ACR), are taken as the competing unpaired training approaches. For LPD and AR, we develop a PyTorch-based implementation based on their publicly available TensorFlow codes[2][3], while for ACR, we use the publicly available PyTorch implementation[4].

The unrolled network $\mathcal{G}_\phi$ has 20 layers, with $5 \times 5$ filters in both primal and dual spaces to increase the overall receptive field for sparse-view measurements. The hyper-parameters involved in training the

---

[2]LPD: `https://github.com/adler-j/learned_primal_dual`.

[3]AR: `https://github.com/lunz-s/DeepAdverserialRegulariser`.

[4]ACR: `https://github.com/Subhadip-1/data_driven_convex_regularization`.

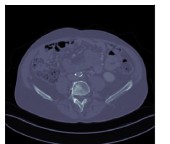 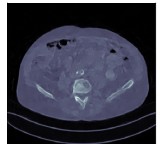 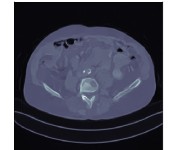 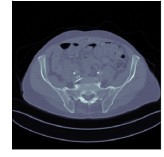 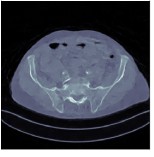 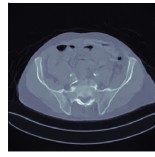

(a) ground-truth    (b) 34.94, 0.88    (c) 35.46, 0.90    (d) ground-truth    (e) 33.84, 0.87    (f) 34.24, 0.89

Figure 3: Effect of refinement: (b) and (e): end-to-end reconstruction $\mathcal{G}_{\phi^*}(\boldsymbol{y}^{\delta})$; (c) and (f): the respective refined reconstructions. The PSNR (dB) and SSIM scores are indicated below.

UAR are specified in Algorithm 1. We found that training a baseline regularizer and a corresponding baseline reconstruction operator stabilized the training process. Training UAR took approximately three hours per epoch on an NVIDIA Quadro RTX 6000 GPU (with 24 GB of memory).

The average performance on the test images in terms of PSNR and SSIM [35] indicates that UAR (with $\lambda = 0.1$ and no refinement) outperforms AR and ACR by approximately 0.5 dB and 2.8 dB, respectively (see Table 1). We would like to emphasize that this gain was found to be consistent across all test images and not just realized on average. With the refinement step, UAR surpasses AR by almost 1 dB and slightly surpasses the U-net-based post-processing. The end-to-end UAR reconstruction is a couple of orders of magnitude faster than AR, while the reduction in reconstruction time is by a factor of four with the refinement. Since the refinement step entails running a few gradient-descent iterations on a high-dimensional variational loss, it makes UAR slower than supervised end-to-end methods. However, thanks to a superior initialization provided by the generator, one requires significantly fewer gradient-descent iterations to refine as compared to a fully variational scheme such as AR. The reconstructions of a representative test image using the competing methods are shown in Fig. 1 for a visual comparison. The effect of $\lambda$ on the reconstruction of UAR is demonstrated in Fig. 2, which confirms the theoretical results in Section 3.2. The refinement step also visibly improves the reconstruction quality of the end-to-end operator, as shown in Fig. 3.

Some additional numerical examples for the task of CT reconstruction, and some illustrative examples of the performance of UAR on two other important imaging inverse problems, namely inpainting and denoising, are provided in Sec. B of the supplementary document.

## 5 Conclusions and outlook

To the best of our knowledge, this work makes the first attempt to blend end-to-end reconstruction with data-driven regularization via an adversarial learning framework. The proposed UAR approach retains the fast reconstruction of the former together with provable guarantees of the latter. We rigorously analyze the proposed framework in terms of well-posedness, noise-stability, and the effect of the regularization penalty, and establish a link between the trained reconstruction operator and the corresponding variational objective. We show strong numerical evidence of the efficacy of the UAR approach for CT reconstruction, wherein it achieves the same performance as supervised data-driven post-processing and outperforms competing unsupervised techniques. Our work paves the way to better understand the role of adversarially learned regularizers in solving ill-posed inverse problems, although several important aspects need further investigation. Since the learned regularizer is non-convex, the performance of gradient-descent on the variational objective greatly depends on initialization. This problem is partly addressed by the unrolled reconstruction operator that efficiently computes a better initial point for gradient descent. However, the precise relationship between the end-to-end reconstruction and the variational minimizer for a given measurement vector remains elusive. Moreover, the quality of the reconstruction relies on the expressive power of neural networks and thus suffers from the *curse of dimensionality*. We believe that addressing such limitations will be important to better understand adversarial regularization methods.

## 6 Acknowledgments

MC acknowledges support from the Royal Society (Newton International Fellowship NIF\R1\192048 Minimal partitions as a robustness boost for neural network classifiers). CBS acknowledges sup-

port from the Philip Leverhulme Prize, the Royal Society Wolfson Fellowship, the EPSRC grants EP/S026045/1 and EP/T003553/1, EP/N014588/1, EP/T017961/1, the Wellcome Innovator Award RG98755, the Leverhulme Trust project Unveiling the invisible, the European Union Horizon 2020 research and innovation programme under the Marie Skodowska-Curie grant agreement No. 777826 NoMADS, the Cantab Capital Institute for the Mathematics of Information, and the Alan Turing Institute.

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
