# End-to-end reconstruction meets data-driven regularization for inverse problems: Supplementary material

**Subhadip Mukherjee**[*1], **Marcello Carioni**[*1], **Ozan Öktem**[2], **and Carola-Bibiane Schönlieb**[1]
[1]Department of Applied Mathematics and Theoretical Physics, University of Cambridge, UK
[2]Department of Mathematics, KTH–Royal institute of Technology, Sweden
[*]Equal contribution authors
Emails: {sm2467, mc2250, cbs31}@cam.ac.uk, ozan@kth.se

The supplementary material consists of this document, which contains the proofs of the theoretical results in Section 3 of the paper and additional experimental results on low-dose CT reconstruction, image inpainting (on MNIST) and denoising (on STL-10). The purpose of these last two experiments is to demonstrate that the proposed UAR framework is applicable to inverse problems in general and is not restricted to CT reconstruction. The data (in `numpy` format) used in the experiments, our `Python` codes (which use the `PyTorch` library for network training), and the instructions for running them are available at `https://github.com/Subhadip-1/unrolling_meets_data_driven_regularization`.

We recall the dominated convergence theorem below, which is used as one of the main tools in our proofs. For the sake of completeness, we also recall the definition of narrow convergence of measures.

**Dominated convergence theorem**: Consider a sequence of measurable functions $\{f_n\}_{n \in \mathbb{N}}$ defined on a measure space $(\Omega, \mathcal{F}, \mu)$ such that $f_n \to f$ pointwise for a measurable function $f$ defined on $(\Omega, \mathcal{F}, \mu)$. Suppose that for any $x \in \Omega$, $|f_n(x)| \leq g(x)$, where $\int_\Omega |g| \, \mathrm{d}\mu < \infty$. Then, it holds that

$$\lim_{n \to \infty} \int_\Omega |f_n - f| \, \mathrm{d}\mu = 0$$

and consequently, $\lim_{n \to \infty} \int_\Omega f_n \, \mathrm{d}\mu = \int_\Omega f \, \mathrm{d}\mu$.

**Narrow convergence of measures**: Consider a sequence of measures $\{\mu_n\}_{n \in \mathbb{N}}$ defined on a measurable space $(\Omega, \mathcal{F})$. Given a measure $\mu$ defined on $(\Omega, \mathcal{F})$ we say that $\mu_n$ narrowly converges to $\mu$ if

$$\lim_{n \to +\infty} \int \varphi \, \mathrm{d}\mu_n = \int \varphi \, \mathrm{d}\mu$$

for every $\varphi \in C_b(\Omega)$, where we denote by $C_b(\Omega)$ the set of bounded continuous functions on $\Omega$.

## A Proofs of the theoretical results

In this section, we prove the theoretical results stated in Section 3. First, we recall the setting and the main definitions. For the set of assumptions used in this section, we refer to Assumptions A1 – A4 stated in Section 3. The objective of the adversarial optimization is defined as

$$\inf_{\phi} \sup_{\mathcal{R} \in \mathbb{L}_1} J_1 \left( \mathcal{G}_\phi, \mathcal{R} | \lambda, \pi_{y^\delta} \right) := \mathbb{E}_{\pi_{y^\delta}} \left\| \boldsymbol{y}^\delta - \mathcal{A} \, \mathcal{G}_\phi(\boldsymbol{y}^\delta) \right\|_2^2 + \lambda \left( \mathbb{E}_{\pi_{y^\delta}} \left[ \mathcal{R}(\mathcal{G}_\phi(\boldsymbol{y}^\delta)) \right] - \mathbb{E}_{\pi_x} \left[ \mathcal{R}(\boldsymbol{x}) \right] \right).$$
(1)

35th Conference on Neural Information Processing Systems (NeurIPS 2021).

In Section 3, we claimed that the problem (1) is well-posed and is equivalent to

$$\inf_{\phi} J_2 \left( \mathcal{G}_{\phi} | \lambda, \pi_{y^{\delta}} \right) := \mathbb{E}_{\pi_{y^{\delta}}} \left\| \boldsymbol{y}^{\delta} - \mathcal{A} \, \mathcal{G}_{\phi}(\boldsymbol{y}^{\delta}) \right\|_2^2 + \lambda \, \mathbb{W}_1(\pi_x, (\mathcal{G}_{\phi})_{\#} \pi_{y^{\delta}}). \tag{2}$$

This shows the connection between the training objective and the Wasserstein-1 distance between the ground-truth distribution and the distribution of the reconstruction. Here, we prove the theorems stated in Section 3 regarding well-posedness (Theorem 1), stability to noise (Theorem 2), and dependence on the parameter $\lambda$ (Proposition 1, Theorem 3, and Theorem 4) for (1) and (2). Moreover, we further discuss the relation between (2) and the variational problem used as a refinement and prove Proposition 2.

## A.1 Well-posedness of the adversarial loss: Proofs of Theorem 1 and Theorem 2

### A.1.1 Proof of Theorem 1

We start by proving the existence of an optimal solution for (2). Let $\mathcal{G}_{\phi_n}$ be a minimizing sequence for (2), namely a sequence of reconstruction operators such that

$$\lim_{n \to +\infty} J_2 \left( \mathcal{G}_{\phi_n} | \lambda, \pi_{y^{\delta}} \right) = \inf_{\phi} J_2 \left( \mathcal{G}_{\phi} | \lambda, \pi_{y^{\delta}} \right). \tag{3}$$

As $\phi_n \in K$ and $K$ is compact and finite dimensional (see Assumption A2), there exists $\phi^* \in K$ such that, up to sub-sequences, $\phi_n \to \phi^*$ and consequently $\mathcal{G}_{\phi_n} \to \mathcal{G}_{\phi^*}$ pointwise (see Assumption A3). We now show that $\mathcal{G}_{\phi^*}$ is a minimum for (2). Thanks to the continuity of $\mathcal{A}$, we know that $\left\| \boldsymbol{y}^{\delta} - \mathcal{A} \, \mathcal{G}_{\phi_n}(\boldsymbol{y}^{\delta}) \right\|_2^2 \to \left\| \boldsymbol{y}^{\delta} - \mathcal{A} \, \mathcal{G}_{\phi^*}(\boldsymbol{y}^{\delta}) \right\|_2^2$ pointwise. Moreover, using Assumptions A1 and A4, the bound

$$\left\| \boldsymbol{y}^{\delta} - \mathcal{A} \, \mathcal{G}_{\phi_n}(\boldsymbol{y}^{\delta}) \right\|_2^2 \leq \sup_{\boldsymbol{y}^{\delta} \in \operatorname{supp}(\pi_{y^{\delta}})} 2 \| \boldsymbol{y}^{\delta} \|_2^2 + 2 \| \mathcal{A} \|_{\mathrm{op}}^2 \left( \sup_{\phi \in K} \| \mathcal{G}_{\phi} \|_{\infty} \right)^2 < \infty$$

holds for every $\boldsymbol{y}^{\delta} \in \operatorname{supp}(\pi_{y^{\delta}})$, where we denote by $\| \mathcal{A} \|_{\mathrm{op}}$ the operator norm of $\mathcal{A}$. Therefore, by applying the dominated convergence theorem, we obtain that

$$\lim_{n \to +\infty} \mathbb{E}_{\pi_{y^{\delta}}} \left\| \boldsymbol{y}^{\delta} - \mathcal{A} \, \mathcal{G}_{\phi_n}(\boldsymbol{y}^{\delta}) \right\|_2^2 = \mathbb{E}_{\pi_{y^{\delta}}} \left\| \boldsymbol{y}^{\delta} - \mathcal{A} \, \mathcal{G}_{\phi^*}(\boldsymbol{y}^{\delta}) \right\|_2^2. \tag{4}$$

Notice now that for every $\varphi \in C_b(\mathbb{R}^k)$.

$$\left| \int \varphi(\boldsymbol{x}) \, \mathrm{d}[(\mathcal{G}_{\phi_n})_{\#} \pi_{y^{\delta}}] - \varphi(\boldsymbol{x}) \, \mathrm{d}[(\mathcal{G}_{\phi^*})_{\#} \pi_{y^{\delta}}] \right| \leq \int \left| \varphi(\mathcal{G}_{\phi_n}(\boldsymbol{y}^{\delta})) - \varphi(\mathcal{G}_{\phi^*}(\boldsymbol{y}^{\delta})) \right| \, \mathrm{d}\pi_{y^{\delta}} \to 0$$

as $n \to +\infty$, using, again, the dominated convergence theorem together with Assumption A1. Thus, the probability measures $(\mathcal{G}_{\phi_n})_{\#} \pi_{y^{\delta}}$ converge narrowly to $(\mathcal{G}_{\phi^*})_{\#} \pi_{y^{\delta}}$ as $n \to +\infty$. Moreover, using again dominated convergence, together with the bound $\sup_{\phi \in K} \| \mathcal{G}_{\phi} \|_{\infty} < \infty$ (see Assumption A4), we also have

$$\left| \int \| \boldsymbol{x} \|_2 \, \mathrm{d}(\mathcal{G}_{\phi_n})_{\#} \pi_{y^{\delta}} - \| \boldsymbol{x} \|_2 \, \mathrm{d}(\mathcal{G}_{\phi^*})_{\#} \pi_{y^{\delta}} \right| \leq \int \left| \| \mathcal{G}_{\phi_n}(\boldsymbol{y}^{\delta}) \|_2 - \| \mathcal{G}_{\phi^*}(\boldsymbol{y}^{\delta}) \|_2 \right| \, \mathrm{d}\pi_{y^{\delta}} \to 0 \tag{5}$$

as $n \to +\infty$. Thus, using [4, Theorem 5.11], we infer that $\lim_{n \to +\infty} \mathbb{W}_1(\pi_x, (\mathcal{G}_{\phi_n})_{\#} \pi_{y^{\delta}}) = \mathbb{W}_1(\pi_x, (\mathcal{G}_{\phi^*})_{\#} \pi_{y^{\delta}})$. Finally using such convergence, together with (4) and (3), we conclude that

$$\begin{aligned}
J_2 \left( \mathcal{G}_{\phi^*} | \lambda, \pi_{y^{\delta}} \right) &= \mathbb{E}_{\pi_{y^{\delta}}} \left\| \boldsymbol{y}^{\delta} - \mathcal{A} \, \mathcal{G}_{\phi^*}(\boldsymbol{y}^{\delta}) \right\|_2^2 + \lambda \, \mathbb{W}_1(\pi_x, (\mathcal{G}_{\phi^*})_{\#} \pi_{y^{\delta}}) \\
&= \lim_{n \to +\infty} \mathbb{E}_{\pi_{y^{\delta}}} \left\| \boldsymbol{y}^{\delta} - \mathcal{A} \, \mathcal{G}_{\phi_n}(\boldsymbol{y}^{\delta}) \right\|_2^2 + \lambda \, \mathbb{W}_1(\pi_x, (\mathcal{G}_{\phi_n})_{\#} \pi_{y^{\delta}}) \\
&= \lim_{n \to +\infty} J_2 \left( \mathcal{G}_{\phi_n} | \lambda, \pi_{y^{\delta}} \right) \\
&= \inf_{\phi} J_2 \left( \mathcal{G}_{\phi} | \lambda, \pi_{y^{\delta}} \right),
\end{aligned}$$

thus showing that $\mathcal{G}_{\phi^*}$ is a minimum for (2).

We now show that (1) and (2) are equivalent. Using the Kantorovich-Rubinstein duality [4, Theorem 1.39] and bound $\sup_{\phi \in K} \|\mathcal{G}_\phi\|_\infty < \infty$ (Assumption A4), we have that for every $\phi \in K$, there exists $\mathcal{R}^\phi \in \mathbb{L}_1$ such that

$$\mathcal{R}^\phi \in \arg\max_{\mathcal{R} \in \mathbb{L}_1} \left( \mathbb{E}_{\pi_{y^\delta}} \left[ \mathcal{R}(\mathcal{G}_\phi(y^\delta)) \right] - \mathbb{E}_{\pi_x} \left[ \mathcal{R}(x) \right] \right), \text{ and} \tag{6}$$

$$\mathbb{E}_{\pi_{y^\delta}} \left[ \mathcal{R}^\phi(\mathcal{G}_\phi(y^\delta)) \right] - \mathbb{E}_{\pi_x} \left[ \mathcal{R}^\phi(x) \right] = \mathbb{W}_1(\pi_x, (\mathcal{G}_\phi)_\# \pi_{y^\delta}). \tag{7}$$

Therefore, denoting by $\mathcal{G}_{\phi^*}$ the minimum for (2), it holds that

$$\inf_\phi \sup_{\mathcal{R} \in \mathbb{L}_1} J_1 \left( \mathcal{G}_\phi, \mathcal{R} | \lambda, \pi_{y^\delta} \right)$$

$$= \inf_\phi \mathbb{E}_{\pi_{y^\delta}} \left\| y^\delta - \mathcal{A} \mathcal{G}_\phi(y^\delta) \right\|_2^2 + \lambda \sup_{\mathcal{R} \in \mathbb{L}_1} \left( \mathbb{E}_{\pi_{y^\delta}} \left[ \mathcal{R}(\mathcal{G}_\phi(y^\delta)) \right] - \mathbb{E}_{\pi_x} \left[ \mathcal{R}(x) \right] \right)$$

$$= \inf_\phi \mathbb{E}_{\pi_{y^\delta}} \left\| y^\delta - \mathcal{A} \mathcal{G}_\phi(y^\delta) \right\|_2^2 + \lambda \left( \mathbb{E}_{\pi_{y^\delta}} \left[ \mathcal{R}^\phi(\mathcal{G}_\phi(y^\delta)) \right] - \mathbb{E}_{\pi_x} \left[ \mathcal{R}^\phi(x) \right] \right)$$

$$= \inf_\phi \mathbb{E}_{\pi_{y^\delta}} \left\| y^\delta - \mathcal{A} \mathcal{G}_\phi(y^\delta) \right\|_2^2 + \lambda \mathbb{W}_1(\pi_x, (\mathcal{G}_\phi)_\# \pi_{y^\delta})$$

$$= \mathbb{E}_{\pi_{y^\delta}} \left\| y^\delta - \mathcal{A} \mathcal{G}_{\phi^*}(y^\delta) \right\|_2^2 + \lambda \left( \mathbb{E}_{\pi_{y^\delta}} \left[ \mathcal{R}^*(\mathcal{G}_{\phi^*}(y^\delta)) \right] - \mathbb{E}_{\pi_x} \left[ \mathcal{R}^*(x) \right] \right),$$

where $\mathcal{R}^*$ is any 1-Lipschitz function such that

$$\mathcal{R}^* \in \arg\max_{\mathcal{R} \in \mathbb{L}_1} \left( \mathbb{E}_{\pi_{y^\delta}} \left[ \mathcal{R}(\mathcal{G}_{\phi^*}(y^\delta)) \right] - \mathbb{E}_{\pi_x} \left[ \mathcal{R}(x) \right] \right).$$

In particular, (11) in Theorem 1 holds and the pair $(\mathcal{G}_{\phi^*}, \mathcal{R}^*)$ is optimal for (1). Viceversa, if $(\mathcal{G}_{\phi^*}, \mathcal{R}^*)$ is optimal for (1), then for every $\hat\phi \in K$ we have

$$J_2 \left( \mathcal{G}_{\phi^*} | \lambda, \pi_{y^\delta} \right) = \sup_{\mathcal{R} \in \mathbb{L}_1} J_1 \left( \mathcal{G}_{\phi^*}, \mathcal{R} | \lambda, \pi_{y^\delta} \right) = \inf_\phi \sup_{\mathcal{R} \in \mathbb{L}_1} J_1 \left( \mathcal{G}_\phi, \mathcal{R} | \lambda, \pi_{y^\delta} \right)$$

$$\leq \sup_{\mathcal{R} \in \mathbb{L}_1} J_1 \left( \mathcal{G}_{\hat\phi}, \mathcal{R} | \lambda, \pi_{y^\delta} \right) = J_2 \left( \mathcal{G}_{\hat\phi} | \lambda, \pi_{y^\delta} \right)$$

where we used the optimality of $(\mathcal{G}_{\phi^*}, \mathcal{R}^*)$ together with (6) and (7), showing that $\mathcal{G}_{\phi^*}$ is a minimizer for (2). $\qquad\square$

### A.1.2   Proof of Theorem 2

Let $\delta_n$ be a sequence converging to $\delta$ as $n \to +\infty$ and

$$\phi_n \in \arg\inf_\phi J_2 \left( \mathcal{G}_\phi | \lambda, \pi_{y^{\delta_n}} \right). \tag{8}$$

Recall that $\pi_{y^{\delta_n}}$ converges in total variation to $\pi_{y^\delta}$. We denote this convergence by

$$\lim_{n \to +\infty} \| \pi_{y^{\delta_n}} - \pi_{y^\delta} \|_\mathcal{M} = 0. \tag{9}$$

Using the fact that $\phi_n \in K$ and $K$ is compact and finite dimensional, we know that there exists $\phi^* \in K$ such that $\phi_n \to \phi^*$, up to sub-sequences. In particular, by Assumption A3, $\mathcal{G}_{\phi_n} \to \mathcal{G}_{\phi^*}$, up to sub-sequences. We need to prove that

$$\phi^* \in \arg\min_\phi J_2 \left( \mathcal{G}_\phi | \lambda, \pi_{y^\delta} \right). \tag{10}$$

First, notice that as $\pi_{y^{\delta_n}} \to \pi_{y^\delta}$ in total variation, it holds that for every bounded, measurable function $f$,

$$\int f \, d\pi_{y^{\delta_n}} \to \int f \, d\pi_{y^\delta}. \tag{11}$$

Therefore, using the fact that $\mathcal{G}_\phi$ is bounded for every $\phi \in K$ (Assumption A4), $\mathcal{A}$ is linear, and the supports of $\pi_{y_n^\delta}$ and $\pi_{y^\delta}$ are uniformly contained in a common compact set (Assumption A1), it holds for every $\mathcal{G}_\phi$ that

$$\lim_{n \to +\infty} \int \| y^\delta - \mathcal{A} \mathcal{G}_\phi(y^\delta) \|_2^2 \, d\pi_{y^{\delta_n}} = \int \| y^\delta - \mathcal{A} \mathcal{G}_\phi(y^\delta) \|_2^2 \, d\pi_{y^\delta}. \tag{12}$$

Moreover, thanks to the dominated convergence theorem, together with the pointwise convergence $\mathcal{G}_{\phi_n} \to \mathcal{G}_{\phi^*}$ and the uniform bound $\sup_n \|\mathcal{G}_{\phi_n}\|_\infty < \infty$ (Assumption A4), we have

$$\lim_{n \to +\infty} \int \|\boldsymbol{y}^\delta - \mathcal{A}\,\mathcal{G}_{\phi_n}(\boldsymbol{y}^\delta)\|_2^2 \, \mathrm{d}\pi_{y^\delta} = \int \|\boldsymbol{y}^\delta - \mathcal{A}\,\mathcal{G}_{\phi^*}(\boldsymbol{y}^\delta)\|_2^2 \, \mathrm{d}\pi_{y^\delta} . \tag{13}$$

Therefore

$$\limsup_{n \to +\infty} \left| \int \|\boldsymbol{y}^\delta - \mathcal{A}\,\mathcal{G}_{\phi_n}(\boldsymbol{y}^\delta)\|_2^2 \, \mathrm{d}\pi_{y^{\delta_n}} - \int \|\boldsymbol{y}^\delta - \mathcal{A}\,\mathcal{G}_{\phi^*}(\boldsymbol{y}^\delta)\|_2^2 \, \mathrm{d}\pi_{y^\delta} \right|$$

$$= \limsup_{n \to +\infty} \left| \int \|\boldsymbol{y}^\delta - \mathcal{A}\,\mathcal{G}_{\phi_n}(\boldsymbol{y}^\delta)\|_2^2 \, \mathrm{d}(\pi_{y^{\delta_n}} - \pi_{y^\delta} + \pi_{y^\delta}) - \int \|\boldsymbol{y}^\delta - \mathcal{A}\,\mathcal{G}_{\phi^*}(\boldsymbol{y}^\delta)\|_2^2 \, \mathrm{d}\pi_{y^\delta} \right|$$

$$\leq \limsup_{n \to +\infty} \left| \int \|y - \mathcal{A}\,\mathcal{G}_{\phi_n}(\boldsymbol{y}^\delta)\|_2^2 \, \mathrm{d}\pi_{y^\delta} - \int \|y - \mathcal{A}\,\mathcal{G}_{\phi^*}(\boldsymbol{y}^\delta)\|_2^2 \, \mathrm{d}\pi_{y^\delta} \right|$$

$$+ \left| \int \|\boldsymbol{y}^\delta - \mathcal{A}\,\mathcal{G}_{\phi_n}(\boldsymbol{y}^\delta)\|_2^2 \, \mathrm{d}(\pi_{y^\delta} - \pi_{y^{\delta_n}}) \right|$$

$$= \limsup_{n \to +\infty} \left| \int \|\boldsymbol{y}^\delta - \mathcal{A}\,\mathcal{G}_{\phi_n}(\boldsymbol{y}^\delta)\|_2^2 \, \mathrm{d}(\pi_{y^\delta} - \pi_{y^{\delta_n}}) \right| \tag{14}$$

$$\leq \limsup_{n \to +\infty} \int 2\|\boldsymbol{y}^\delta\|_2^2 + 2(\|\mathcal{A}\|_{\mathrm{op}} \sup_n \|\mathcal{G}_{\phi_n}\|_\infty)^2 \, \mathrm{d}\|\pi_{y^\delta} - \pi_{y^{\delta_n}}\| \tag{15}$$

$$\leq \left[ \sup_{\boldsymbol{y}^\delta \in \mathcal{K}} 2\|\boldsymbol{y}^\delta\|_2^2 + 2\|\mathcal{A}\|_{\mathrm{op}}^2 \Big( \sup_{\phi \in K} \|\mathcal{G}_\phi\|_\infty \Big)^2 \right] \limsup_{n \to +\infty} \|\pi_{y^\delta} - \pi_{y^{\delta_n}}\|_{\mathcal{M}} = 0, \tag{16}$$

where in (14) we use (13) and in (15)–(16) we use (9) together with the fact that $\mathcal{G}_{\phi_n}$ are uniformly bounded (Assumption A4), $\mathcal{A}$ is linear and the supports of $\pi_{y_n^\delta}$ and $\pi_{y^\delta}$ are uniformly contained in a common compact set $\mathcal{K}$ (Assumption A1).

Consider now a test function $\varphi \in C_b(\mathbb{R}^k)$. Notice that

$$\limsup_{n \to +\infty} \left| \int \varphi(\mathcal{G}_{\phi_n}(\boldsymbol{y}^\delta)) \, \mathrm{d}\pi_{y^{\delta_n}} - \int \varphi(\mathcal{G}_{\phi^*}(\boldsymbol{y}^\delta)) \, \mathrm{d}\pi_{y^\delta} \right|$$

$$\leq \limsup_{n \to +\infty} \left| \int \varphi(\mathcal{G}_{\phi_n}(\boldsymbol{y}^\delta)) \, \mathrm{d}(\pi_{y^{\delta_n}} - \pi_{y^\delta}) \right| + \left| \int \varphi(\mathcal{G}_{\phi_n}(\boldsymbol{y}^\delta)) \, \mathrm{d}\pi_{y^\delta} - \int \varphi(\mathcal{G}_{\phi^*}(\boldsymbol{y}^\delta)) \, \mathrm{d}\pi_{y^\delta} \right|$$

$$\leq \limsup_{n \to +\infty} \int |\varphi(\mathcal{G}_{\phi_n}(\boldsymbol{y}^\delta))| \, \mathrm{d}|\pi_{y^{\delta_n}} - \pi_{y^\delta}| + \left| \int \varphi(\mathcal{G}_{\phi_n}(y)) \, \mathrm{d}\pi_{y^\delta} - \int \varphi(\mathcal{G}_{\phi^*}(\boldsymbol{y}^\delta)) \, \mathrm{d}\pi_{y^\delta} \right|$$

$$\leq \limsup_{n \to +\infty} \|\varphi\|_\infty \|\pi_{y^{\delta_n}} - \pi_{y^\delta}\|_{\mathcal{M}} + \left| \int \varphi(\mathcal{G}_{\phi_n}(\boldsymbol{y}^\delta)) \, \mathrm{d}\pi_{y^\delta} - \int \varphi(\mathcal{G}_{\phi^*}(\boldsymbol{y}^\delta)) \, \mathrm{d}\pi_{y^\delta} \right|$$

$$= 0,$$

where we use again (9) together with the pointwise convergence $\mathcal{G}_{\phi_n} \to \mathcal{G}_{\phi^*}$ and the compactness of the support of $\pi_{y^\delta}$. Such estimate prove that $(\mathcal{G}_{\phi_n})_\# \pi_{y^{\delta_n}}$ converges narrowly to $(\mathcal{G}_{\phi^*})_\# \pi_{y^\delta}$. Moreover, adapting the previous to test function $\varphi(\boldsymbol{x}) = \|\boldsymbol{x}\|_2$ and using additionally that $\sup_n \|\mathcal{G}_{\phi_n}\|_\infty < \infty$ (see Assumption A4) we infer

$$\lim_{n \to +\infty} \left| \int \|\boldsymbol{x}\|_2 \, \mathrm{d}[(\mathcal{G}_{\phi_n})_\# \pi_{y^{\delta_n}}] - \|\boldsymbol{x}\|_2 \, \mathrm{d}[(\mathcal{G}_{\phi^*})_\# \pi_{y^\delta}] \right| = 0$$

which, thanks to [4, Theorem 5.11] and together with the narrow convergence $(\mathcal{G}_{\phi_n})_\# \pi_{y^{\delta_n}} \to (\mathcal{G}_{\phi^*})_\# \pi_{y^\delta}$ implies

$$\lim_{n \to +\infty} \mathbb{W}_1(\pi_x, (\mathcal{G}_{\phi_n})_\# \pi_{y^{\delta_n}}) = \mathbb{W}_1(\pi_x, (\mathcal{G}_{\phi^*})_\# \pi_{y^\delta}) \tag{17}$$

and similarly

$$\lim_{n \to +\infty} \mathbb{W}_1(\pi_x, (\mathcal{G}_\phi)_\# \pi_{y^{\delta_n}}) = \mathbb{W}_1(\pi_x, (\mathcal{G}_\phi)_\# \pi_{y^\delta}) \quad \text{for all } \phi \in K. \tag{18}$$

We are finally in position to prove (10). Let $\phi \in K$ a competitor for the variational problem in (10). Then thanks to the optimality of $\phi_n$

$$J_2\left(\mathcal{G}_{\phi_n}|\lambda, \pi_{y^{\delta_n}}\right) \leq J_2\left(\mathcal{G}_\phi|\lambda, \pi_{y^{\delta_n}}\right)$$

for every $n$. Passing to the limit in the previous inequality using (17), (18), (12) and (16) we obtain

$$J_2\left(\mathcal{G}_{\phi^*}|\lambda, \pi_{y^\delta}\right) \leq J_2\left(\mathcal{G}_\phi|\lambda, \pi_{y^\delta}\right) \tag{19}$$

as we wanted to prove. $\qquad\square$

## A.2 Effect of $\lambda$ on the end-to-end reconstruction. Proofs of Proposition 1, Theorem 3 and Theorem 4

Here we prove Proposition 1, Theorem 3 and Theorem 4. We remind the reader the definition of the function spaces

$$\Phi_\mathcal{L} := \left\{\phi : \mathbb{E}_{\pi_{y^\delta}} \left\|\boldsymbol{y}^\delta - \mathcal{A}\,\mathcal{G}_\phi(\boldsymbol{y}^\delta)\right\|_2^2 = 0\right\} \text{ and } \Phi_\mathbb{W} := \left\{\phi : (\mathcal{G}_\phi)_{\#}\pi_{y^\delta} = \pi_x\right\}$$

that we assume to be non-empty.

### A.2.1 Proof of Proposition 1

Let $\mathcal{G}_{\phi^*}$ be a minimizer for (2). Then for every $\phi \in \Phi_\mathcal{L}$ we easily estimate

$$\mathbb{E}_{\pi_{y^\delta}} \left\|\boldsymbol{y}^\delta - \mathcal{A}\,\mathcal{G}_{\phi^*}(\boldsymbol{y}^\delta)\right\|_2^2$$
$$\leq \mathbb{E}_{\pi_{y^\delta}} \left\|\boldsymbol{y}^\delta - \mathcal{A}\,\mathcal{G}_\phi(\boldsymbol{y}^\delta)\right\|_2^2 + \lambda\,\mathbb{W}_1(\pi_x, (\mathcal{G}_\phi)_{\#}\pi_{y^\delta}) - \lambda\,\mathbb{W}_1(\pi_x, (\mathcal{G}_{\phi^*})_{\#}\pi_{y^\delta})$$
$$\leq \lambda\,\mathbb{W}_1(\pi_x, (\mathcal{G}_\phi)_{\#}\pi_{y^\delta}),$$

leading to the first estimate in Proposition 1. Moreover, for every $\phi \in \Phi_\mathbb{W}$ we obtain the second estimate in Proposition 1, that is

$$\lambda\mathbb{W}_1(\pi_x, (\mathcal{G}_{\phi^*})_{\#}\pi_{y^\delta})$$
$$\leq \mathbb{E}_{\pi_{y^\delta}} \left\|\boldsymbol{y}^\delta - \mathcal{A}\,\mathcal{G}_\phi(\boldsymbol{y}^\delta)\right\|_2^2 + \lambda\,\mathbb{W}_1(\pi_x, (\mathcal{G}_\phi)_{\#}\pi_{y^\delta}) - \mathbb{E}_{\pi_{y^\delta}} \left\|\boldsymbol{y}^\delta - \mathcal{A}\,\mathcal{G}(\boldsymbol{y}^\delta)\right\|_2^2$$
$$\leq \mathbb{E}_{\pi_{y^\delta}} \left\|\boldsymbol{y}^\delta - \mathcal{A}\,\mathcal{G}_\phi(\boldsymbol{y}^\delta)\right\|_2^2,$$

where we used that for every $\phi \in \Phi_\mathbb{W}$, $\mathbb{W}_1(\pi_x, (\mathcal{G}_\phi)_{\#}\pi_{y^\delta}) = 0$. $\qquad\square$

### A.2.2 Proof of Theorem 3

We are assuming $\lambda_n \to 0$ and

$$\phi'_n \in \arg\inf_\phi J_2\left(\mathcal{G}_\phi|\lambda_n, \pi_{y^\delta}\right). \tag{20}$$

First, using the fact that $\phi'_n \in K$ and $K$ is compact and finite dimensional we know that there exists $\phi_1^* \in K$ such that $\phi'_n \to \phi_1^*$, up to sub-sequences. In particular, it also holds that $\mathcal{G}_{\phi'_n} \to \mathcal{G}_{\phi_1^*}$, up to sub-sequences, by Assumption A3. It remains to prove that

$$\phi_1^* \in \arg\min_{\phi \in \Phi_\mathcal{L}} \mathbb{W}_1(\pi_x, (\mathcal{G}_\phi)_{\#}\pi_{y^\delta}). \tag{21}$$

First notice that by Proposition 1 we can select $\phi \in \Phi_\mathcal{L}$ such that

$$\mathbb{E}_{\pi_{y^\delta}} \left\|\boldsymbol{y}^\delta - \mathcal{A}\,\mathcal{G}_{\phi'_n}(\boldsymbol{y}^\delta)\right\|_2^2 \leq \lambda_n\mathbb{W}(\pi_x, (\mathcal{G}_\phi)_{\#}\pi_{y^\delta})$$

for every $n$. So, taking the limit for $n \to +\infty$ and using that $\lambda_n \to 0$ together with (4) (where again we used Assumptions A1 and A4, and the dominated convergence theorem) we obtain $\mathbb{E}_{\pi_{y^\delta}} \left\|\boldsymbol{y}^\delta - \mathcal{A}\,\mathcal{G}_{\phi_1^*}(\boldsymbol{y}^\delta)\right\|_2^2 = 0$. Now, let $\phi \in \Phi_\mathcal{L}$. Using (20) we have that for every $n$

$$\lambda_n\mathbb{W}_1(\pi_x, (\mathcal{G}_{\phi'_n})_{\#}\pi_{y^\delta}) \leq \lambda_n\mathbb{W}_1(\pi_x, (\mathcal{G}_{\phi'_n})_{\#}\pi_{y^\delta}) + \mathbb{E}_{\pi_{y^\delta}} \left\|\boldsymbol{y}^\delta - \mathcal{A}\,\mathcal{G}_{\phi'_n}(\boldsymbol{y}^\delta)\right\|_2^2$$
$$\leq \lambda_n\mathbb{W}_1(\pi_x, (\mathcal{G}_\phi)_{\#}\pi_{y^\delta}). \tag{22}$$

With similar arguments as in the proof of Theorem 1 we can prove that the probability measures $(\mathcal{G}_{\phi'_n})_{\#}\pi_{y^\delta}$ converge narrowly to $(\mathcal{G}_{\phi_1^*})_{\#}\pi_{y^\delta}$ as $n \to +\infty$. Additionally using the bound $\sup_{\phi \in K} \|\mathcal{G}_\phi\|_\infty < \infty$ (see Assumption A4) we can repeat the computation in (5) to prove that

$\lim_{n \to +\infty} \mathbb{W}_1(\pi_x, (\mathcal{G}_{\phi'_n})_{\#}\pi_{y^\delta}) = \mathbb{W}_1(\pi_x, (\mathcal{G}_{\phi_1^*})_{\#}\pi_{y^\delta})$ [4, Theorem 5.11]. So, passing to the limit in (22) we conclude that

$$\mathbb{W}_1(\pi_x, (\mathcal{G}_{\phi_1^*})_{\#}\pi_{y^\delta}) = \lim_{n \to +\infty} \mathbb{W}_1(\pi_x, (\mathcal{G}_{\phi'_n})_{\#}\pi_{y^\delta}) \leq \mathbb{W}_1(\pi_x, (\mathcal{G}_\phi)_{\#}\pi_{y^\delta})$$

showing (21). We now prove the convergence $\lim_{n \to \infty} \frac{1}{\lambda_n} \inf_\phi J_2\left(\mathcal{G}_\phi | \lambda_n, \pi_{y^\delta}\right) = \mathbb{W}_1(\pi_x, (\mathcal{G}_{\phi_1^*})_{\#}\pi_{y^\delta})$.

Notice that using that, as $\mathbb{E}_{\pi_{y^\delta}} \left\| y^\delta - \mathcal{A}\,\mathcal{G}_{\phi_1^*}(y^\delta) \right\|_2^2 = 0$ and (20) we have

$$\frac{1}{\lambda_n} J_2\left(\mathcal{G}_{\phi'_n} | \lambda_n, \pi_{y^\delta}\right) \leq \mathbb{W}_1(\pi_x, (\mathcal{G}_{\phi_1^*})_{\#}\pi_{y^\delta})$$

and trivially

$$\frac{1}{\lambda_n} J_2\left(\mathcal{G}_{\phi'_n} | \lambda_n, \pi_{y^\delta}\right) \geq \mathbb{W}_1(\pi_x, (\mathcal{G}_{\phi'_n})_{\#}\pi_{y^\delta}).$$

So, passing to the limit in the previous estimates and using that $\lim_{n \to +\infty} \mathbb{W}_1(\pi_x, (\mathcal{G}_{\phi'_n})_{\#}\pi_{y^\delta}) = \mathbb{W}_1(\pi_x, (\mathcal{G}_{\phi_1^*})_{\#}\pi_{y^\delta})$ we prove the desired convergence. $\qquad\square$

### A.2.3 Proof of Theorem 4

We are assuming $\lambda_n \to +\infty$ and

$$\phi'_n \in \arg\inf_\phi J_2\left(\mathcal{G}_\phi | \lambda_n, \pi_{y^\delta}\right). \tag{23}$$

First, using the fact that $\phi'_n \in K$ and $K$ is compact and finite dimensional we know that there exists $\phi_2^* \in K$ such that $\phi'_n \to \phi_2^*$, up to sub-sequences. In particular, it also holds that $\mathcal{G}_{\phi'_n} \to \mathcal{G}_{\phi_2^*}$, up to sub-sequences, by Assumption A3. It remains to prove that

$$\phi_2^* \in \arg\min_{\phi \in \Phi_\mathbb{W}} \mathbb{E}_{\pi_{y^\delta}} \left\| y^\delta - \mathcal{A}\,\mathcal{G}_\phi(y^\delta) \right\|_2^2. \tag{24}$$

First notice that by Proposition 1 we can select $\phi \in \Phi_\mathbb{W}$ such that

$$\mathbb{W}_1(\pi_x, (\mathcal{G}_{\phi'_n})_{\#}\pi_{y^\delta}) \leq \frac{\mathbb{E}_{\pi_{y^\delta}} \left\| y^\delta - \mathcal{A}\,\mathcal{G}_\phi(y^\delta) \right\|_2^2}{\lambda_n} \tag{25}$$

for every $n$. With similar arguments as in the proof of Theorem 1, using Assumption A1 and Assumption A4 together with [4, Theorem 5.11] there holds that $\lim_{n \to +\infty} \mathbb{W}_1(\pi_x, (\mathcal{G}_{\phi'_n})_{\#}\pi_{y^\delta}) = \mathbb{W}_1(\pi_x, (\mathcal{G}_{\phi_2^*})_{\#}\pi_{y^\delta})$. So, taking the limit in (25) for $n \to +\infty$ and using that $\lambda_n \to +\infty$ we obtain $\mathbb{W}_1(\pi_x, (\mathcal{G}_{\phi_2^*})_{\#}\pi_{y^\delta}) = 0$.

Let now $\phi \in \Phi_\mathbb{W}$. Using (23) we have that for every $n$

$$\mathbb{E}_{\pi_{y^\delta}} \left\| y^\delta - \mathcal{A}\,\mathcal{G}_{\phi'_n}(y^\delta) \right\|_2^2 \leq \mathbb{E}_{\pi_{y^\delta}} \left\| y^\delta - \mathcal{A}\,\mathcal{G}_{\phi'_n}(y^\delta) \right\|_2^2 + \lambda_n \mathbb{W}_1(\pi_x, (\mathcal{G}_{\phi'_n})_{\#}\pi_{y^\delta})$$
$$\leq \mathbb{E}_{\pi_{y^\delta}} \left\| y^\delta - \mathcal{A}\,\mathcal{G}_\phi(y^\delta) \right\|_2^2. \tag{26}$$

With similar arguments as in the proof of Theorem 1, using dominated convergence theorem together with Assumption A1 and Assumption A4 it holds that

$$\lim_{n \to +\infty} \mathbb{E}_{\pi_{y^\delta}} \left\| y^\delta - \mathcal{A}\,\mathcal{G}_{\phi'_n}(y^\delta) \right\|_2^2 = \mathbb{E}_{\pi_{y^\delta}} \left\| y^\delta - \mathcal{A}\,\mathcal{G}_{\phi_2^*}(y^\delta) \right\|_2^2. \tag{27}$$

So, passing to the limit in (26) we conclude that

$$\mathbb{E}_{\pi_{y^\delta}} \left\| y^\delta - \mathcal{A}\,\mathcal{G}_{\phi_2^*}(y^\delta) \right\|_2^2 = \lim_{n \to +\infty} \mathbb{E}_{\pi_{y^\delta}} \left\| y^\delta - \mathcal{A}\,\mathcal{G}_{\phi'_n}(y^\delta) \right\|_2^2 \leq \mathbb{E}_{\pi_{y^\delta}} \left\| y^\delta - \mathcal{A}\,\mathcal{G}_\phi(y^\delta) \right\|_2^2$$

showing (24).

We now show the convergence $\lim_{n \to \infty} \inf_\phi J_2\left(\mathcal{G}_\phi | \lambda_n, \pi_{y^\delta}\right) = \mathbb{E}_{\pi_{y^\delta}} \left\| y^\delta - \mathcal{A}\,\mathcal{G}_{\phi_2^*}(y^\delta) \right\|_2^2$. Using that, $\mathbb{W}_1(\pi_x, (\mathcal{G}_{\phi_2^*})_{\#}\pi_{y^\delta}) = 0$, together with (23) we have

$$J_2\left(\mathcal{G}_{\phi'_n} | \lambda_n, \pi_{y^\delta}\right) \leq \mathbb{E}_{\pi_{y^\delta}} \left\| y^\delta - \mathcal{A}\,\mathcal{G}_{\phi_2^*}(y^\delta) \right\|_2^2$$

and trivially

$$J_2\left(\mathcal{G}_{\phi'_n} | \lambda_n, \pi_{y^\delta}\right) \geq \mathbb{E}_{\pi_{y^\delta}} \left\| y^\delta - \mathcal{A}\,\mathcal{G}_{\phi'_n}(y^\delta) \right\|_2^2.$$

So, passing to the limit in the previous estimate and using (27) we prove the desired convergence. $\quad\square$

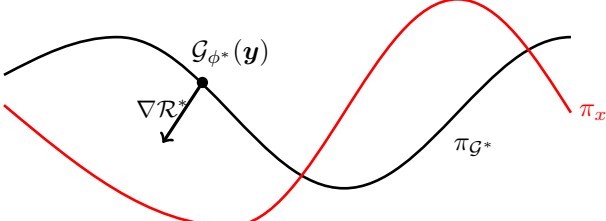

Figure 1: A step of gradient descent applied to the initial point $\mathcal{G}_{\phi^*}(\boldsymbol{y})$ moves the point in the direction $\nabla\mathcal{R}^*(\mathcal{G}_{\phi^*}(\boldsymbol{y}))$ closer to the data distribution $\pi_x$.

### A.3 End-to-end reconstruction vis-à-vis the variational solution. Proof of Proposition 2 and further discussion

#### A.3.1 Proof of Proposition 2

The first upper bound in Proposition 2 is a simple application of Markov inequality for probability measures, which states that every non-negative random variable $U$ satisfies $\mathbb{P}\left(U \geq \eta\right) \leq \frac{\mathbb{E}[U]}{\eta}$, for any $\eta > 0$.

For the second upper bound notice that using Theorem 1 we have

$$\int \mathcal{R}^*(\boldsymbol{x})\,\mathrm{d}[(\mathcal{G}_{\phi^*})_\#\pi_{y^\delta}] = \int \mathcal{R}^*(\boldsymbol{x})\,\mathrm{d}[(\mathcal{G}_{\phi^*})_\#\pi_{y^\delta}] - \int \mathcal{R}^*(\boldsymbol{x})\,\mathrm{d}\pi_x = \mathbb{W}_1(\pi_x, (\mathcal{G}_{\phi^*})_\#\pi_{y^\delta}), \tag{28}$$

where we also use the assumption $\mathcal{R}^*(\boldsymbol{x}) = 0$ for $\pi_x$-almost every $\boldsymbol{x}$. Therefore the second upper bound in Proposition 2 follows from an application of Markov inequality, thanks to the assumed positivity of $\mathcal{R}^*$. $\qquad\square$

We remark the assumption regarding the positivity of $\mathcal{R}^*$ is not restrictive, as $\mathcal{R}^* + C$ is optimal for every $C \in \mathbb{R}$. However, it is not always true that $\mathcal{R}^*(\boldsymbol{x}) = 0$ for $\pi_x$-almost every $\boldsymbol{x}$. As discussed in Section 3, such assumption can be justified using a suitable weak manifold assumption for $\pi_x$.

We conclude this section by further discussing the content of Theorem 5. As already noted, this theorem ensures that a gradient-descent step performed on $\mathcal{R}^*$ at $\boldsymbol{x} = \mathcal{G}_{\phi^*}(\boldsymbol{y}^\delta)$ decreases the Wasserstein distance with respect to the ground-truth distribution $\pi_x$. Therefore, if the gradient-descent step to solve the variational problem (8) is initialized with the reconstruction $\mathcal{G}_{\phi^*}(\boldsymbol{y}^\delta)$, the next iterate gets pushed closer to the ground-truth distribution $\pi_x$. If we additionally use the same weak manifold assumption as in [2] it is possible to prove that an optimal regularizer $\mathcal{R}^*$ is given by the distance function from the ground-truth manifold (see [2]). In this case, if we additionally assume that the projection from $\mathcal{G}_{\phi^*}(\boldsymbol{y}^\delta)$ to the manifold is unique, then the gradient of $\mathcal{R}^*$ in that point is a unit vector from $\mathcal{G}_{\phi^*}(\boldsymbol{y}^\delta)$ to the unique projection point. Such consideration strengthens even more our claim that an iterate of gradient descent initialized in $\mathcal{G}_{\phi^*}(\boldsymbol{y}^\delta)$ gets pushed closer to the ground-truth distribution $\pi_x$. A graphical representation of such effect is presented in Fig. 1.

## B Additional numerical results

### B.1 Some additional CT reconstruction examples

First, we provide a comparison of different algorithms on another test image from the Mayo-clinic low-dose CT challenge dataset [3] (See Figure 2 below). The purpose of this example is to demonstrate that the gain in performance achieved by UAR over the competing algorithms is consistent over different test images, and is not just on average over all the test images.

### B.2 Illustrative examples for inpainting and denoising

In this section, we consider two important imaging inverse problems: (i) image inpainting and (ii) denoising; and show some representative examples demonstrating the performance of UAR for these

tasks. The inpainting experiment is conducted on the MNIST dataset, where the measurement has a square 8x8 block of missing pixels in the middle and is additionally corrupted by Gaussian noise with variance equal to 0.2 (See Figure 3 below). UAR does a reasonable job of reconstructing the underlying true digits. The UAR generator and regularizer are trained on 60000 training images and then evaluated on the remaining 10000 test images for inpainting. The training batch-size is 128 and the models are trained for 25 epochs. The regularization penalty is taken to be 0.2 for this experiment. The generator is an unrolled proximal gradient network with 20 layers, which also goes on to show that the UAR scheme is independent of the particular choice of the optimization algorithm that is unrolled to construct the generator. The refinement was computed by running 50 iterations of the variational problem starting from the end-to-end reconstruction as the initial estimate. We noted that the refinement led to only minor improvements in the MSE. The average PSNR (dB) and SSIM scores over the test images are as follows: (i) measurement: 12.21 dB, 0.42; (ii) UAR (end-to-end): 22.12 dB, SSIM 0.91; and (iii) UAR (refined): 22.17 dB, 0.91.

The denoising experiment on the STL-10 dataset is conducted on images corrupted by Gaussian noise of standard deviation 0.1 (see Figure 4). For this experiment as well, we noted that the refinement resulted in little to no further improvement on the initial end-to-end reconstruction. The average PSNR (dB) and SSIM over the 800 test images are as follows: (i) noisy measurement: 19.99 dB, 0.91; (ii) UAR (end-to-end and refined) 25.23 dB, 0.97.

These experiments demonstrate that the proposed UAR framework is applicable to imaging inverse problems in general, although the training and network hyper-parameters are to be carefully engineered to extract the state-of-the-art performance from UAR for a specific inverse problem of interest. In both inpainting and denoising experiments, we chose to unroll the proximal gradient-descent algorithm (as opposed to PDHG [1] that was chosen for the CT experiments), which go on to show that the UAR framework is not specific to the choice of the algorithm that one unfolds to construct the generator. For both inpainting and denoising experiments, we found that the refinement step did not yield noticeable improvement over the initial estimate obtained by the end-to-end generator. This is due to the fact that both MNIST and STL-10 datasets are not too diverse/heterogeneous and there are 60000 and 5000 training images, respectively, which is much higher than the number of available training images in the Mayo-CT dataset that we used for the CT experiment. Consequently, the end-to-end generator, which learns to minimize the expected variational loss over the distribution of the measurement, already provides a reconstruction that is very close to the true minimizer of the variational objective for a specific given measurement. This trend indicates that as the number of training images increases, the improvement from the refinement step tends to diminish and one already gets a reasonably good reconstruction by just using the end-to-end unrolled generator.


Figure 2: Another numerical example on the Mayo clinic data [3]. As we see, UAR (refined) significantly outperforms AR and ACR, and achieves slightly better reconstruction quality than U-net-based post-processing, which is a supervised approach. To see the reduction in reconstruction time using UAR as compared to competing variational methods (such as TV, AR, and ACR), refer to Table 1 in the main manuscript.

    (b) Did you specify all the training details (e.g., data splits, hyperparameters, how they were chosen)? [Yes] Please refer to Section 4 and Algorithm 1.

    (c) Did you report error bars (e.g., with respect to the random seed after running experiments multiple times)? [Yes] Please refer to Section 4 and Table 1.

    (d) Did you include the total amount of compute and the type of resources used (e.g., type of GPUs, internal cluster, or cloud provider)? [Yes] See Section 4.

4. If you are using existing assets (e.g., code, data, models) or curating/releasing new assets...

    (a) If your work uses existing assets, did you cite the creators? [Yes] Please refer to Section 4 for references to the dataset and the codes used in this work.

    (b) Did you mention the license of the assets? [Yes] The dataset was made public by Mayo-Clinic [3].

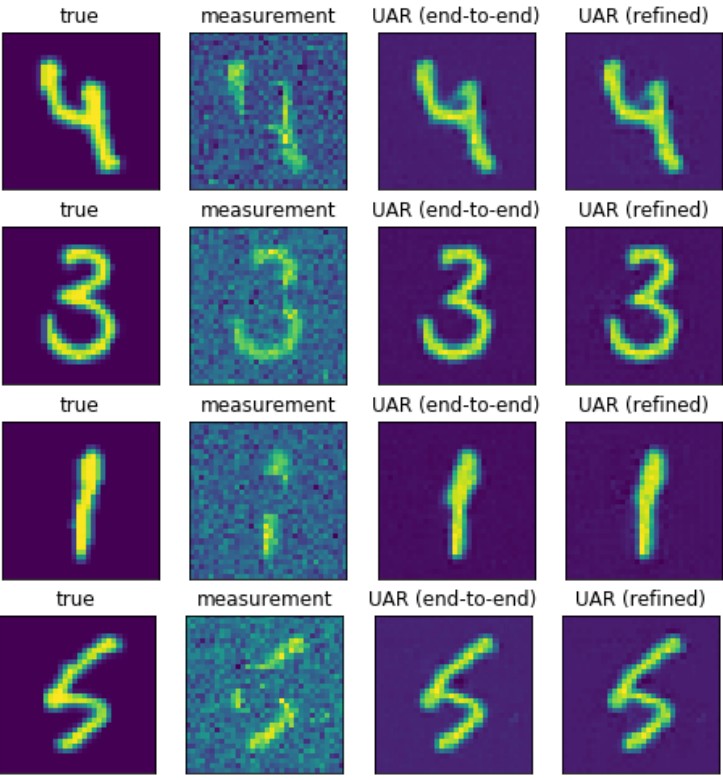

Figure 3: Some representative examples of inpainting MNIST digits.

(c) Did you include any new assets either in the supplemental material or as a URL? [Yes] Our code is included in the supplemental material.

(d) Did you discuss whether and how consent was obtained from people whose data you're using/curating? [Yes] The dataset and the codes used in this work are available publicly.

(e) Did you discuss whether the data you are using/curating contains personally identifiable information or offensive content? [Yes] In Section 4, we mention that the data used in this work are publicly available [3]. For the ease of running our scripts, we convert the raw data to numpy format, which does not contain any sensitive information, and make them available along with the supplementary material.

5. If you used crowdsourcing or conducted research with human subjects...

(a) Did you include the full text of instructions given to participants and screenshots, if applicable? [N/A]

(b) Did you describe any potential participant risks, with links to Institutional Review Board (IRB) approvals, if applicable? [N/A]

(c) Did you include the estimated hourly wage paid to participants and the total amount spent on participant compensation? [N/A]

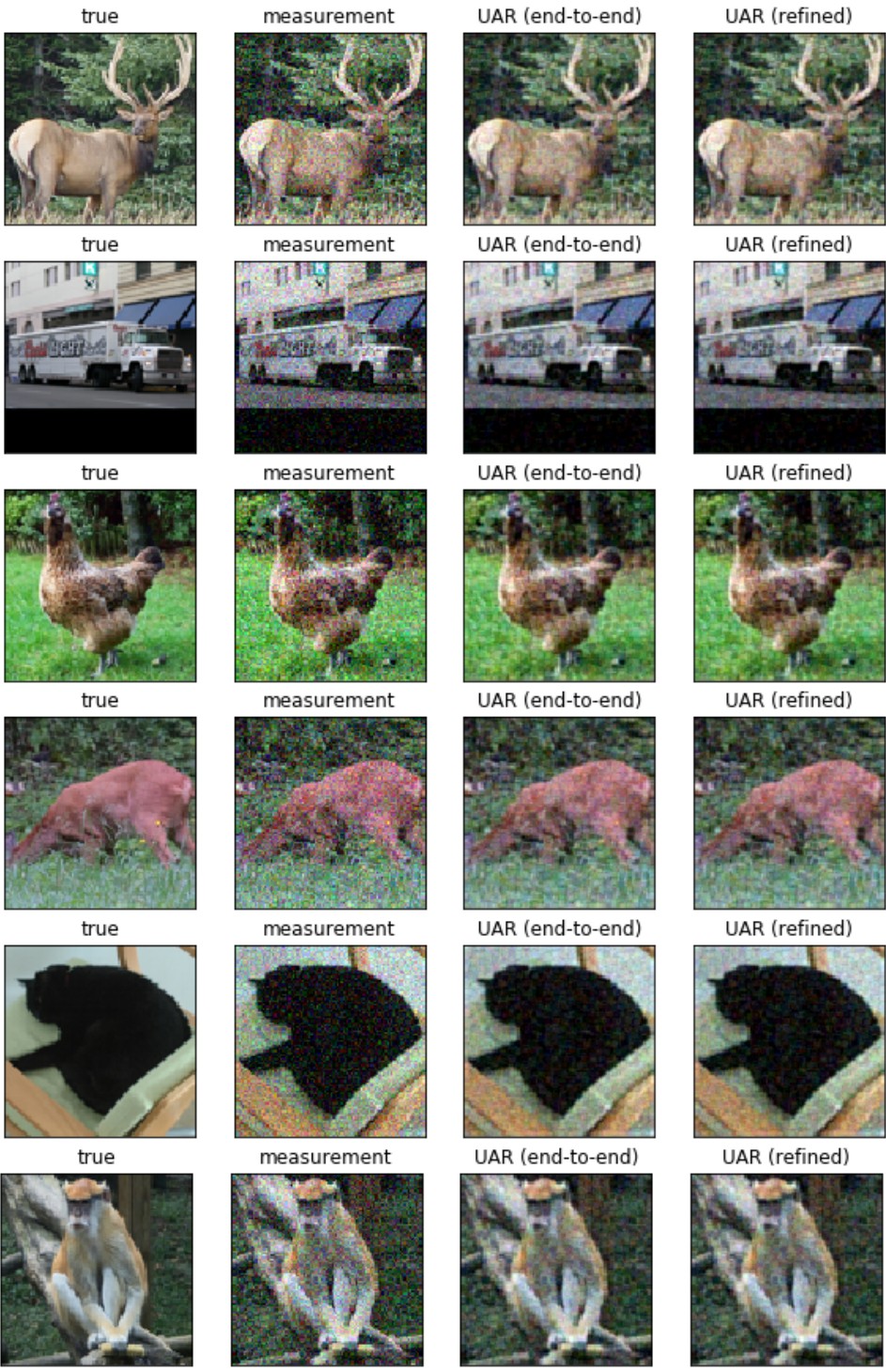

Figure 4: Some representative examples of denoising on STL-10.