# OpenReview forum: "End-to-end reconstruction meets data-driven regularization for inverse problems"
_NeurIPS.cc/2021/Conference — NeurIPS 2021 Poster_

### Official Review · Reviewer_EFMB · 2021-07-16

**Rating:** 6
**Confidence:** 3

**Summary:**

This paper proposes a framework, named UAR, for jointly training an iterative deep unfolding neural network (DUNN) and an adversarial discriminator based on Wasserstein distance for solving image inverse problems, inspired by adversarial regularization (AR) method (Lunz’2018). Unlike original AR method that takes pseudo-inverse reconstruction as input images, the proposed method assimilates the reconstruction from DUNN that unrolls the iterative primal-dual algorithm in each training step. The authors also provide theoretical analysis about when such adversarial training with unrolling networks would be successful. The theoretical results are well established by with thorough proofs. Finally, the performance of the proposed method is evaluated for solving CT inverse problem with satisfactory results compared to both supervised and unsupervised deep learning methods such as U-Net, LPD, and AR.

**Limitations And Societal Impact:**

**Weakness: ** My main concerns with this paper are as follows.

- The claim about “proposing an unsupervised approach” seems to be misleading for me. The proposed method still needs to partially take ground truth images as input in Algorithm 1 for every training steps. Similarly, As pointed out by the authors presented line 109 – 116, “the role of $R_\theta$ is to discern ground-truth images”, the proposed method is actually not 100% ground truth free.

- The theoretical results are assumed that the maximization in $R$ is performed over the space of _1-lipz_ functions. However, the 1-lipz constrain loss presented in Algorithm 1 seems not to be a strong constrain, which may weaken the theoretical claims.

-  While the authors give a brief overview of PnP and RED in the background, the comparison with those advanced iterative methods is missing. When equipped with advanced denoisers even trained on nature images, those methods can still achieve excellent performance on various invers image problems [R2, R3]. A related method[R4] that uses artifact remove CNN as regularizer trained without ground truth achieves state-of-the-art performance on real MRI data.  Hence, the authors should give them more credits and at least discuss them in the related works to make the paper more comprehensive.

[R1] Deep unfolding network for image super-resolution

[R2] SIMBA: Scalable Inversion in Optical Tomography using Deep Denoising Priors.

[R3] Plug-and-Play Methods Provably Converge with Properly Trained Denoisers

[R4] Rare: image reconstruction using deep priors learned without ground truth

**Main Review:**

**Originality:** This work essentially proposes  a technic for training DUNNs with adversarial networks using Wasserstein loss, inspired by previous AR method.  Apart from AR, the idea of training DUNNs with adversarial loss is also demonstrated by recent work such as USRNet (R1) that using relativistic adversarial loss to train DUNN for image super-resolution. Hence, the methodical contributions are limited. However, this work also builds up interesting theoretical analysis about when such adversarial training for unrolling networks would be successful. The theoretical results are well established with thorough proofs and are validated by numerical results shown in Table 1 and Figure 2. Although these analyses are not radically novel, it could have broad impact on the field that it could enable adversarial training for DUNNs.

**Quality & Clarity:**  Overall the paper is  fairly well written and easy to read. However, I found it hard to follow at some specific sections due to the insufficient details on the notations.  For one, the definition and meaning of $\delta$ in $\pi_{y^\delta} $ that is first presented in line 118 is not clearly stated. Similarly, it would be better to introduce the 1-lipschitz constrained loss in Algorithm 1 with more details. On the other hand, some typos exist such as missing $k$ for $\phi^k$ in Eq. 4.

**Significance:** I think this is a well written paper, in a line of research that is interesting and should be of interest to the NeurIPS community.


**Time Spent Reviewing:**

4

---

> ### Author Response · Authors · 2021-08-09
> **Response to Reviewer EFMB**
>
> Thanks for your assessment of the paper. Please find below the responses to the concerns that you have raised.
>
> 1. On quality and clarity: We use $\delta$ to denote the norm of the additive noise in the measurement. Please refer to Page-1, Line-21 where $\delta$ is defined. The probability distribution of the noisy measurement $\boldsymbol{y}^{\delta}$ is denoted by $\pi_{\boldsymbol{y}^{\delta}}$. Imposing the 1-Lipschitz constraint using a gradient-penalty is a standard approach in the Wasserstein GAN literature. See Proposition 1 in "Improved Training of Wasserstein GANs" by Gulrajani et al. for further details on this (https://arxiv.org/pdf/1704.00028.pdf). We will include this reference in the revised paper. Finally, we don't think there is a typo in eq. (4). To clarify, $J_k^{(1)}(\phi)$ is a function of the generator parameter $\phi$ for a given regularizer parameter $\theta_k$. Minimizing $J_k^{(1)}(\phi)$ with respect to $\phi$ gives the updated generator parameter $\phi_k$.
>
> 2. Yes, we agree that the proposed approach is not ground-truth-free. We are willing to call the training approach "unpaired" to avoid any confusion.
>
> 3. It is indeed true that the 1-Lipschitz constraint is imposed via a soft penalty. However, the gradient-norm of the resulting learned regularizer turns out to be not so far from 1 in practice. Moreover, the theoretical results do not assume that $\mathcal{R}$ has to be strictly 1-Lipschitz. As long as $\mathcal{R}$ is $L$-Lipschitz for some finite $L$, one can rewrite the regularization term as $\lambda L\frac{\mathcal{R}(\boldsymbol{x})}{L}$, and interpret $\lambda L$ as the new penalty and think of $\frac{\mathcal{R}(\boldsymbol{x})}{L}$ as a 1-Lipschitz regularizer. The theoretical claims still hold true in this case.
>
> 4. PnP and RED approaches use pre-trained denoisers as image priors to regularize inverse problems, building on the idea that well-trained denoisers contain information about the data distribution. On the contrary, the regularizer $R_\theta$ and the end-to-end reconstruction $G_\phi$ in our approach are learned simultaneously by an adversarial learning framework. Consequently, the learned regularizer is specific for the end-to-end reconstruction, leading to higher quality reconstructions. The idea of learning a data-driven regularizer and a custom generator network simultaneously is the main contribution of the paper (along with the theoretical results on its connections to optimal transport), which we believe is substantially different from PnP and RED. We compared UAR to AR, which we considered as a representative algorithm for the class of data-driven regularization (which PnP and RED are also members of). Nevertheless, we will add a more nuanced discussion on the links between our approach with PnP and RED in the revised paper.

---

> > ### Comment · Reviewer_EFMB · 2021-08-26
> > **Keeping my current score rating**
> >
> > Thank you for taking the time to respond to my and other reviewers’ comments.
> >
> > I believe the authors have thoroughly responded my comments. The term of supervised/unsupervised in the revision/responses is clarified. The additional numerical results will make this paper even stronger.  Although the proposed design builds upon existing ideas and concepts, based on a previously proposed AR and Wasserstein GANs, many new deep neural network designs can be seen as incremental developments of the older ones, yet they are needed for the progress of the field. Similarly, I do think the work is significant in the sense that getting a better theoretical understanding and control of the well-working approaches that utilize deep unfolding with variational methods is important.
> >
> > Hence, I would like to maintain my score for now.

---

### Official Review · Reviewer_meHG · 2021-07-16

**Rating:** 7
**Confidence:** 4

**Summary:**

This paper proposes a method for adversarial regularization, to be used in image reconstruction. This method relies on a joint training of the regularizing part of the system and the reconstruction. Several properties regarding convergence are proven, and the method is shown to outperform similar regularization methods for this task.

**Limitations And Societal Impact:**

The authors have described limiations of their work in the final part of the paper, in that the reconstruction quality depends on the architectures used, as well as some limitations on the relationship of the losses used for given samples.

As mentioned by the authors, this method is used for the well-known problem of image reconstuction, so I agree that further analysis of potential negative social impact is not required.

**Main Review:**

This paper builds on a previously proposed Adversarial Regularization (AR) method. AR involves learning a regularizer network, which attempts to discriminate images coming from the ground truth or the reconstructed distribution. In this work, the authors propose UAR, which goes one step further, in that the regularizer is jointly trained with the reconstructing network, in an adversarial fashion. This is a significant advancement from the AR method.

In its current form, UAR seems to share a lot of similarities with Wasserstein GANs, and thus parallels can be drawn with other works which have used similar techniques for image reconstruction. For example, there is some work on using a similar technique to UAR on MR images (Lei et al., 2020). Even though I think comparison with such works should be included, I think the originality of this paper is not significantly hampered, since the authors also include theoretical statements and proofs on the properties of UAR.

The authors describe several theoretical properties of UAR, such as stability of optimal point to noise, behavior based on varying the strength of the regularization, and probabilistic bounds on the reconstruction error. I believe that the existence of these is useful to fully understand the properties of UAR, and also a major part of the contribution of the paper.

The experimental section of the paper focuses on the problem of CT images. The proposed method is compared with both supervised and unsupervised approaches, and it is shown that it is capable of outperforming the AR and ACR unsupervised methods. I find it particularly interesting that that UAR is 2 orders of magnitude faster than AR, even though intuitively it would make sense to me that AR should converge faster (since the regularizer and the reconstruction are trained separately). I believe that this result merits further discussion in the experimental section.

The paper is clearly written and easy to follow. I have some small comments I want to make:
- In Algorithm 1, it seems that the batch size is set as equal to 1. I am not sure if this is correct or a typo, because it seems quite a weird choice to make for batch size.
- Again in Algorithm 1, the way for loops are written implies looping over both mini-batches and epochs. I think it would be clearer if the epochs were written separately.
- Section 2.2.2 is used to define the reconstruction network $G_\phi$, but I think the way it is placed breaks the flow of the paper a little, because Section 2.2.3 refers to a refinement step, which I believe would be better to mention directly after the original training process in Section 2.2.1. I would suggest either moving Section 2.2.2, or including its contents in the previous one.
- There is a small typo in line 143, for the definition of the reconstruction network.

Overall, I believe this is a good and interesting submission. I have some slight reservations about the originality of the method proposed, but I think these are overcome by the theoretical analysis provided.

References:

K. Lei, M. Mardani, J. M. Pauly and S. S. Vasanawala, "Wasserstein GANs for MR Imaging: From Paired to Unpaired Training," in IEEE Transactions on Medical Imaging, vol. 40, no. 1, pp. 105-115, Jan. 2021.

**Post-rebuttal comment**:

After the author response, I have decided to keep my score (see below comment for details).

**Time Spent Reviewing:**

8

---

> ### Author Response · Authors · 2021-08-09
> **Response to Reviewer meHG**
>
> Thank you for your assessment of the paper. The proposed UAR method indeed relies on optimal transport (and hence Wasserstein GANs) and combines it with the adversarial learning framework for data-adaptive regularizers. Please find below the specific responses to the concerns that you have raised.
>
> 1. UAR vs. AR in terms of convergence speed: In UAR, the generator $G_{\phi^*}$ is trained to minimize (on average) the variational objective with $R_{\theta^*}$ as the regularizer. Consequently, the generator output $G_{\phi^*}({y}^{\delta})$ provides a really good initial solution for the variational problem with the UAR regularizer $R_{\theta^*}$ for any given measurement $y^{\delta}$. Thanks to the superior initial point, the UAR refinement converges in much fewer gradient-descent iterations than AR. Nevertheless, the training of UAR is computationally more expensive than AR, since it involves learning two networks adversarially instead of just a regularizer. We will make this difference more clear in the revised paper.
>
>
> 2. The batch-size is indeed 1, which is primarily because of limited GPU memory. With the image and projection size chosen for the experiment, along with the network sizes, we could not fit a larger batch-size on the GPU. This isn't particularly limiting in our opinion, since we were able to achieve reconstruction quality that is competitive with, and in some cases superior to the state-of-the-art approaches. One can increase the batch-size (and possibly the reconstruction quality) by using multiple GPUs. Moreover, we would also like to point out that a batch-size of 1 is not a weird choice as compared to the existing literature on training LPD-like schemes. See [3] for example (in Page-6, Column-2, first para), where a batch-size of 1 is used for training LPD (in a supervised manner) on human phantoms.
>
>
> 3. We agree that one can equivalently write the same loop in Algo. 1 using an outer loop over epochs and an inner loop over mini-batches. We will modify this in the revised paper.
>
>
> 4. Thanks for your suggestions on the organization of the subsections. we will rearrange the material in the revised paper as you suggest.
>
>
>
> 5. Thanks for pointing out the typo. The networks in the primal and the dual spaces should be $\Lambda_{\phi_{\text{p}}^{\ell}}$ and $\Lambda_{\phi_{\text{d}}^{\ell}}$, respectively. We will correct it in the revised paper.

---

> > ### Comment · Reviewer_meHG · 2021-08-26
> > **Post-Rebuttal Comment**
> >
> > Thank you very much for your response to both my and the rest of the reviewers' comments.
> >
> > I believe that the paper is strong in its current form, due to its theoretical contributions along with the additional experiments performed during the rebuttal process.
> >
> > I am keeping my score for now, mostly because of me still having some reservations about comparison with similar techniques based on optimal transport / W-GANs (as mentioned in my main review), but as I previously stated I believe that the paper is good even without them, due to the theoretical analysis of the proposed method.

---

### Official Review · Reviewer_RFxg · 2021-07-16

**Rating:** 6
**Confidence:** 3

**Summary:**

This paper proposes a framework for training an adversarial network, where the discriminator is a regularization functional, for image reconstruction.
A similar work mentioned in the paper is [14]. The key difference between [14] and the proposed method, UAR, is that the former is based on the variational inference while the latter replaces it with an unfolding network, making UAR a pure end-to-end training deep learning model.
Since UAR incorporates the forward model in the loss function, it is self-supervised requiring only $y$ to impose supervision.
The author also addresses the theoretical properties of UAR, presenting interesting results on the optimality, influence of $\lambda$, and justification of end-to-end learning & regularization.

**Limitations And Societal Impact:**

1. As mentioned before, the rationale of using unfolding networks is not well discussed in the paper. The author should better explain their motivation.

2. In fact, the UAR model can be viewed as an adversarial network with an unfolding network serving as the generator. In other works, the loss for training unfolding networks are just simple MSE $\frac{1}{N}\sum_{i=1}^{N}||x_i-x_i^\ast||_2^2$ with direct supervision. Will the incorporation of the forward model degenerate the final performance of the network despite the benefit of self-supervision? I assume the theoretical results should also apply to this simpler case?

3. Are the weights of layers of nonlinearity shared across different blocks or independent? If they are shared, can the iterative refinement be incorporated into the network by repeating more blocks in the inference time?



**Main Review:**

**Originality:** This paper is original, providing useful theoretical results on the different aspects of adversarial networks.

**Quality:** The quality of this paper is high on the theoretical side. However, the experiment seems inadequate with only one experiment on CT.

**Clarity:** The paper is well-written. But the justification for the usage of unfolding networks as the 'generator' is somewhat missing. In my opinion, the similarity between the unfolding networks and the variational inference may be the reason.

**Significance:** The results present in the paper is of enough significance for acceptance.

**Time Spent Reviewing:**

4 hours

---

> ### Author Response · Authors · 2021-08-09
> **Response to Reviewer RFxg**
>
> Thank you for your evaluation of the paper. Please see below the responses to your comments and feedback.
>
> 1.  We have now implemented the proposed UAR method on two other inverse problems, namely image inpainting (on MNIST) and denoising (on STL-10), and demonstrated that it does a reasonable job of reconstructing the ground-truth in both cases. Please see the new results at the anonymous github repository: https://github.com/uar-new-results/uar_new_results.
> We would like to emphasize that these experiments only provide a proof-of-concept that UAR is not particularly limited to CT. One can improve the performance of UAR by appropriately optimizing the hyper-parameters associated with training and the network architectures.
> If reviewers and Area Chair agree, we are willing to include these new simulations in the supplementary material.
> Additionally, we would also like to mention that although Table 1 reports improvement in reconstruction quality of UAR over the competing algorithms "on average" over the test images, we found that this trend was consistent over all 128 slices extracted from the test patient. Therefore, we believe that the CT experiment serves as a convincing proof of concept for UAR applied to inverse problems.
>
> &NewLine;
> &NewLine;
>
> 2. The rationale behind choosing an unrolled network for the generator is the fact that we want the generator to be a good approximation to the variational solution. For a wide class of convex variational problems, primal-dual hybrid gradient (PDHG) (see [6]) has been shown to enjoy theoretical guarantees and good numerical performance. Although we are aiming to approximate the solution to a non-convex variational problem, PDHG serves as a motivation for parametrizing the generator, especially in view of the excellent expressive power of such parametrization as demonstrated in [3].
>
> &NewLine;
> &NewLine;
>
> 3. The numerical comparison of our approach with an unrolled LPD network trained against MSE is given in Section 4 (Fig. 1 and Table 1) of the paper and also in the supplementary document. The proposed UAR method is slightly inferior to supervised LPD in terms of PSNR and SSIM, so the deterioration in performance is not significant. Moreover, UAR is more preferable than LPD trained on MSE loss because of at least two reasons: (i) one does not need paired examples to train UAR and (ii) the final reconstruction is consistent with the measured data, thanks to the underlying variational framework. The supervised LPD method, however, is not guaranteed to produce data-consistent solutions. The theoretical links of the proposed method with optimal transport hold true in the variational setting. Therefore, it is not immediately clear if the analysis can be extended to the supervised setting, or what such an extension would even mean.
>
> &NewLine;
> &NewLine;
>
> 4. The nonlinearities across the different layers of the generator are allowed to be different. The reason behind this was to achieve enough expressive power to approximate the variational solution with reasonable accuracy with only a few layers. One can of course tie the parameters across the generator layers and study convergence in the limit as the number of layers $L\rightarrow \infty$, but such an analysis would perhaps need more structure on the regularizer (such as convexity). Nevertheless, doing so would be against the objective of the paper in spirit, which is to compute a cheap solution to the variational reconstruction problem. This is therefore best left for a separate investigation.

---

> > ### Comment · Reviewer_RFxg · 2021-08-15
> > **Keeping my 6 rate**
> >
> > Thank you for your detailed response.
> >
> > 1) I believe it will be nice to include new simulations into the supplement as they provide more comparison.
> >
> > 2) Maybe I was not clear with my second comment. My question is why do we need the forward matrix in the loss when the network itself is already an unfolding network, meaning the data consistency has been already enforced via the network structure. The explanation of such redundancy is what I am asking. One potential explanation is to waive ground-truth images.

---

> > > ### Author Response · Authors · 2021-08-18
> > > **Response to Reviewer RFxg: Round-2**
> > >
> > > 1. Thanks for your suggestion. We will include the new simulation results in the supplementary document.
> > >
> > >
> > > 2. Thank you for the clarification. We agree that one reason for incorporating the fidelity term in the measurement domain is to avoid using paired examples (i.e., to avoid direct supervision). Moreover, some existing literature argues that end-to-end approaches (fully trained, post-processing, or iteratively unfolded networks) trained in a supervised manner do not necessarily enforce data consistency, and hence some information in the measured data might get ignored. See the recent paper by Baguer et al. addressing this issue (Section 3.1, after eq(9)): https://arxiv.org/pdf/2003.04989.pdf. Therefore, although the generator network has the forward operator embedded into it, one also additionally needs the data-fidelity loss to ensure that the reconstruction is data-consistent. We will add clarification on this in the revised paper.
> > >
> > >
> > > 3. To explain a bit more with regards to your second comment in the first review, one can indeed simply train an unfolding network to minimize the MSE loss (direct supervision), but this requires paired data, unlike UAR. We also agree that simpler theoretical results could possibly be obtained in this case, but they would have a different flavor as they would involve the joint distribution on $(\boldsymbol{x},\boldsymbol{y})$ and not just the marginals $\pi_x$ and $\pi_y$.

---

> > > > ### Comment · Reviewer_RFxg · 2021-08-29
> > > > **Worth including Point 3 into the paper**
> > > >
> > > > Sorry for my late response. I think point 3 is a good theoretical explanation for the usage of forward matrix in the loss function. Perhaps a short statement like this will eliminate similar questions from readers.

---

> > > > > ### Author Response · Authors · 2021-08-29
> > > > > **Response to Reviewer RFxg: Round-3**
> > > > >
> > > > > Thank you for the suggestion. We will include a statement to this effect in the revised paper.

---

### Official Review · Reviewer_oCFm · 2021-07-16

**Rating:** 7
**Confidence:** 3

**Summary:**

This paper proposes a deep learning based reconstruction for ill-posed inverse problems that combines ideas from end-to-end supervised training with learned regularization in a variational setting. The method adversarially trains the weights of a deep neural network (used as a prior for regularization) while also learning an end-to-end, unrolled deep network. The unrolled network takes the form of a learned primal-dual algorithm, and the regularizer aims to discern between ground-truth images and images generated by the unrolled network. After training, the end-to-end network can be used for reconstruction alone, or as an initialization for a separate reconstruction that strictly uses the learned regularizer in a variational setting. The authors claim that this combines the best of both worlds, as an end-to-end model can be used for fast reconstruction, while the variational model can be used for refinement (while also being more amenable to analysis). The authors provide theoretical and empirical evidence that their approach is successful and outperforms other state-of-the-art methods.


**Limitations And Societal Impact:**

The authors do not address limitations, though the societal impact is positive as it could lead to reduced radiation dose in routine CT imaging.

**Main Review:**

[Edit - I have updated my score from 6 to 7 following the discussion below]

The proposed approach provides a nice connection between end-to-end methods and variational methods. While there are a number of moving parts, the training is easy to understand and can be flexibly applied to other unrolled architectures and learned regularizers. While not new, the use of adversarial training is an interesting departure from conventional end-to-end methods that are trained with pixel-wise losses. The theoretical analysis is thorough and relatively easy to follow. The submission is clearly written and well-organized.

My main concerns with the paper are (1) the use of the term unsupervised, (2) the disconnect between the end-to-end reconstruction and the initialized variational reconstruction, and (3) the limited number of experiments run:

1. Supervised vs. unsupervised:
While it is true that pairs of (image, measurement) data are not strictly required, the method *does* require ground-truth reference images for training. Therefore I believe it is misleading to consider this unsupervised. For example, the following works do not require any access to reference images:

[1a] O. Senouf, S. Vedula, T. Weiss, A. M. Bronstein, O. Michailovich, M. Zibulevsky, Self-supervised learning of inverse problem solvers in medical imaging, Proc. Medical Image Learning with Less Labels and Imperfect Data, MICCAI 2019.
[1b] Jonathan I Tamir, Stella X Yu, Michael Lustig, Unsupervised deep basis pursuit: Learning inverse problems without ground-truth data, NeurIPS Workshop on Deep Inverse Problems, Vancouver, Canada, December 2019.
[1c] Yaman, B, Hosseini, SAH, Moeller, S, Ellermann, J, Uğurbil, K, Akçakaya, M. Self-supervised learning of physics-guided reconstruction neural networks without fully sampled reference data. Magn Reson Med. 2020; 84: 3172– 3191. https://doi.org/10.1002/mrm.28378.
[1d] Elizabeth K Cole, John M Pauly, Shreyas S Vasanawala, Frank Ong, Unsupervised MRI reconstruction with generative adversarial networks, arXiv preprint arXiv:2008.13065.

In particular, the work in [1d] has a very similar formulation, but replaces the optimization in Eq. 5 with one that only requires access to prospectively under-sampled measurement data. Could this learned regularizer still be used in the variational setting? It would be beneficial if the authors could comment on this.

2. End-to-end vs variational reconstruction:
The authors claim that a big benefit of their approach is the ability to use the end-to-end reconstruction as an initialization in a variational framework, where the regularizer previously learned can be used. While this is interesting, there are some comparative details missing. For example, how would this compare if I used some *other* end-to-end reconstruction (for example, the one in Ref [3]) with a regularizer that was separately trained in an adversarial fashion (for example, the one in Ref [14]). To what extent is it critical that they be coupled? In addition, based on Table 1, the number of iterations is reduced by a factor of about 4, while still slower than end-to-end alone by a factor of over 20. Therefore, it feels misleading to say that "the refinement step involves running a few iterations". How would the variational reconstruction perform (speed and quality) if initialized with a different point, e.g. the pseudo-inverse, or a different end-to-end reconstruction such as the simple U-Net?

3. Experiments.
The experiments are limited to a single dataset consisting of 9 patients used for training, one patient used for testing (it is unclear if Supporting Fig 2 is a different patient or a different slice of the same patient), and no explanation of validation data (was the test set actively used for validation?). To me this is very limiting in terms of empirical evidence. Since the paper title is quite general, it would be interesting to see performance for other inverse problems. I also would like to authors to clarify why they added Gaussian noise to the measurements. Based on the webpage for the low-dose CT challenge, I understand that the challenge organizers added Poisson noise to the projection data.

In addition, I have a few minor comments:
4. The notation for the CNNs in Section 2.2.2. differs between the text and the equations.
5. The captions in Figure 1 do not explain what the numbers represent. I presume it is PSNR and SSIM.
6. Since Fig 2 shows a different slice than the one in Fig 1, it would be helpful to show the ground-truth image.



**Time Spent Reviewing:**

4

---

> ### Author Response · Authors · 2021-08-09
> **Response to Reviewer oCFm**
>
> We thank the reviewer for his/her feedback, comments and suggestions. We address in the next list the points raised.
>
> 1. Yes, we agree that the proposed approach is not completely unsupervised as it is not ground-truth free. We are willing to call the training approach "unpaired" to avoid any confusion.
> Thanks for the suggested references. We will include a discussion about them in the revised paper. Specifically, we want to point out that although [1d] proposes an unsupervised training approach using GANs, it differs substantially from our approach. Unlike UAR, the discriminator in [1d] acts in the measurement domain, and not in the image domain, and it is not rooted in the variational setting. Yes, it solves eq. (5) in the measurement domain, but our approach seeks to solve (4) and (5) in an alternating manner, so the loss function in (5) does not remain static during training.
>
> &NewLine;
> &NewLine;
>
>
> 2. Coupling the variational problem corresponding to the learned regularizer and the initialization provided by the associated learned generator is important. See Theorem 5 in the paper (and Figure 1 and Section A.3 in the supplementary document) for a detailed discussion on this. To answer your specific question, if the AR [14] approach is initialized with a learned primal-dual reconstruction [3], the subsequent gradient-descent iterations can worsen the solution.
> We would like to point out that variational methods with a data-driven prior (such as the AR and the proposed UAR) succeed only when they are initialized appropriately. For example, it is important to initialize AR with the unregularized solution (pseudo-inverse), since those images were used as negative examples to train the regularizer in the first place. More specifically, gradient descent on the AR variational objective might fail to converge if it is initialized with any arbitrary estimate. The output of an arbitrary end-to-end method can be quite far from being data-consistent (i.e., far from being consistent with the measurement) and hence a gradient step on the regularizer can drive the subsequent iterate away from the ground-truth distribution. This phenomenon is true for both AR and UAR.
> The convergence of UAR refinement is faster than AR though, because the generator is already trained to minimize the corresponding variational loss on average (over the measurement distribution) and therefore tends to provide a better initial estimate for a given measurement.
> With the sentence "the refinement step involves running a few iterations" we meant to say that the refinement step in UAR converges in fewer iterations as compared to AR, where one does not have a corresponding end-to-end generator. Of course, if one chooses to do refinement, the overall reconstruction time is higher than end-to-end methods. But, the generator in UAR gives a high-quality reconstruction to begin with, so it may not be needed to refine it further depending on the application. We will make this point clearer in the revised version.
>
> &NewLine;
> &NewLine;
>
>
> 3. All experimental results correspond to slices extracted from the same test patient (which was not used during training). No validation data was used in selecting the hyper-parameters. We instead followed the same setting adopted in reference [3] so far as the hyper-parameters in the network and the optimizers are concerned. Moreover, we did not optimize the regularization penalty $\lambda$ by using a validation set. Instead, we show the performance for different $\lambda$ (and give a theoretical analysis on the impact of $\lambda$). And, no, no part of the test set was used for model validation.
> The real projection data has Poisson noise, but the data-likelihood for Poisson noise is different from the standard $\ell_2^2$ distance and is not therefore amenable to the kind of optimal transport based analysis presented in the paper. Hence, we simulated low-dose projections with Gaussian noise to ensure consistency between the numerics and the theoretical results.
> Yes, we agree that the proposed approach is not limited to CT and is applicable to a broader class of inverse problems. However, we chose to focus on sparse-view CT since it is a prototypical inverse problem for which there are not many unpaired training approaches available. Our goal was to demonstrate the potential of our method, showing that it is able to achieve state-of-art performances on this specific task. However, we understand the referee's concerns regarding the applicability of UAR to different inverse problems. Therefore, we performed experiments on inpainting for MNIST digits and denoising on STL-10 images obtaining reasonable reconstructions (see the anonymous github repository https://github.com/uar-new-results/uar_new_results).
> One can further improve the performance of UAR by appropriately optimizing the hyper-parameters associated with training and the network architectures corresponding to a particular application. If reviewers and Area Chair agree, we are willing to include these new simulations in the supplementary material.
>
> Minor comments
>
> 1. Regarding notations, we use $\phi$ to denote the collection of all learnable parameters  ${{\phi_{\text{p}}^{(\ell)},\phi_{\text{d}}^{(\ell)},\sigma^{(\ell)},\tau^{(\ell)}}}$ for $\ell=0, \ldots, L-1$ in the LPD-based generator. We will clarify it in the revised paper.
>  2. Yes, the captions indicate PSNR (in dB) and SSIM.
>  3. Sure, we will include the ground-truth in Figure 2 in the revised paper.

---

> > ### Comment · Reviewer_oCFm · 2021-08-19
> > **Keeping my score pending additional clarification**
> >
> > Thank you for the reply. I am maintaining my score but I am willing to increase it to a 7 pending the following:
> >
> > 1. Your response and planned modifications are satisfactory.
> > 2. In addition to clarifying in the revised version, please also explicitly indicate that it is not "few" iterations, but rather "fewer as compared to AR alone." This is a subtle but important distinction.
> > 3. I appreciate the addition of MNIST inpainting and STL-10 denoising and I believe it will help strengthen the paper's message. I still take issue with the additive Gaussian noise for CT. It can be highly misleading to butcher an inverse problem in order to make it match theoretical assumptions when the true inverse problem differs, especially in the case where data are taken from a challenge that explicitly points out the noise model. At that point why use CT data at all? You could just as well simulate projections of STL-10 and add Gaussian noise. It will probably give you a similar successful result! It would be highly useful to see how the algorithm performs on the actual, real-world inverse problem, while also acknowledging the mismatch when moving from theory to practice. At a minimum, the paper should mention that (1) the mismatch was done in order to change the inverse problem to match the theory, (2) acknowledge that this is a major limitation of the current work, and (3) discuss the importance of validating the approach in the future before evaluating its applicability to an important application such as low-dose CT.
> >
> > Minor comments:
> > 1. I specifically was referring to the Gamma that shows up at the bottom of page 4 but is a Lambda on line 143.
> > 2. Please include this in the figure
> > 3. Thank you!

---

> > > ### Author Response · Authors · 2021-08-20
> > > **Further clarifications for Reviewer oCFm**
> > >
> > > Thank you so much for your comments and assessment. Please see the specific responses below.
> > > &nbsp;
> > > &nbsp;
> > >
> > > 1. Thank you for the assessment.
> > > &nbsp;
> > > &nbsp;
> > >
> > > 2. Sure, we will mention explicitly in the revised paper that UAR-refinement converges in fewer iterations as compared to AR
> > >     to avoid any ambiguity.
> > > &nbsp;
> > > &nbsp;
> > >
> > > 3. We will include the new simulations (on MNIST and STL-10) in the supplementary document.
> > >     &nbsp;
> > >    &nbsp;
> > >
> > >     We are also willing to clearly mention in the revised paper that the CT numerical experiments are performed on simulated projection data with
> > >     Gaussian noise and they serve as a proof-of-concept only for the theory developed in the paper. We are also willing to acknowledge that this gap between our analysis (theoretical and experimental) and actual real-world CT physics is a shortcoming of the paper, and one should rigorously validate the UAR algorithm in the future on more practical CT data.
> > >     &nbsp;
> > >    &nbsp;
> > >
> > >     Nevertheless, for the sake of clarification, we would like to point out that the precise implementations of the forward and the adjoint operators ($A$ and $A^*$, respectively) corresponding to the quarter-dose simulated data provided by the challenge organizers are not available. Since we can only work with the approximate implementations of $A$ and $A^*$, the distortion in the projection space could be considerably different from any standard noise distribution (Poisson or Gaussian). This is why we chose to use ODL (with astra backend) for simulating the projection data from the normal-dose images. The authors of LPD and AR also use the same ODL-based simulator for computing the CT projection data. While Poisson noise is added to the data for LPD (see Section IV.A.2, reference [3]), the numerical experiments for AR are carried out with additive Gaussian noise (see Section 5.3, reference [14]). Notably, the choice of noise statistics in simulating the noisy projection data is somewhat less important for end-to-end supervised methods (such as LPD) as compared to variational approaches (such as UAR and AR), since the former does not seek to explicitly minimize a variational loss involving a data-fidelity term.
> > > &nbsp;
> > > &nbsp;
> > >
> > >
> > > Following your suggestion, we will make this point very precise in the revised paper and accept it as a limitation that needs to be addressed in the future.
> > > &nbsp;
> > > &nbsp;
> > >
> > > Response to minor comments:
> > > &nbsp;
> > > &nbsp;
> > >
> > > 1. Thank you. We will correct this typo in the revised paper.
> > > &nbsp;
> > > &nbsp;
> > >
> > > 2. Sure, we will include this in the Figure.

---

> > > > ### Comment · Reviewer_oCFm · 2021-08-20
> > > > **Changing to a 7**
> > > >
> > > > Thanks for your response. I will update my score to a 7.
> > > >
> > > > Regarding comment 3, it is exactly this reason why it is important to see the performance when the noise statistics differ, since one of the claims of the paper are that the proposed combined end-to-end + data-driven regularization approach is competitive with end-to-end alone. However it is well-known that end-to-end methods can manage just fine with different noise models, while it is not clear that the proposed method would be flexible enough to handle an inverse problem with a different noise model such as low-dose CT.
> > > >
> > > > In any case I thank the reviewers for addressing my concerns.

---

### Official Review · Reviewer_QSGm · 2021-07-18

**Rating:** 6
**Confidence:** 3

**Summary:**

This paper develops a learning based approach for end-to-end reconstruction of operators for ill-posed problems. In particular it examines the problem of image reconstruction. The main contribution is to combine traditional variational approach with an optimal transport based regularizer.

**Limitations And Societal Impact:**

Some limitations have already been listed in the paper.

**Main Review:**

The paper is well written and address a problem of interest. I have however a couple of concerns:

- First there are two well written past works that overlap too much with this paper. One that has been referenced here (see ref 12) by Erich Kobler et. al. at CVPR 2020 where they cover the variational approach in detail and it's application to CT. The second is the paper

Sim, Byeongsu, et al. "Optimal transport driven CycleGAN for unsupervised learning in inverse problems." SIAM Journal on Imaging Sciences 13.4 (2020): 2281-2306.

The above besides addressing optimal transport approaches for inverse problems, also addresses the adversarial training aspect and further looks at various applications, including CT.

- Second, only CT image reconstruction is demonstrated. probably it works with other modalities as well, but it hasn't been demonstrated in the main body of the paper.

**Time Spent Reviewing:**

3 hours

---

> ### Author Response · Authors · 2021-08-10
> **Response to Reviewer QSGm**
>
> We thank the reviewer for his/her feedback. We address in the next list the points raised.
>
> &NewLine;
> &NewLine;
>
> 1. We thank the reviewer for pointing out the related reference (Sim, Byeongsu, et al. "Optimal transport driven CycleGAN for unsupervised learning in inverse problems." SIAM Journal on Imaging Sciences 13.4 (2020): 2281-2306). We will cite it in our revised version. After considering the paper, however, we believe that their approach does not substantially overlap with ours. The adversarial objective of Sim, Byeongsu, et al. (i.e. the functional K defined in (3.3) and (3.4) in the paper) penalizes the Wasserstein distance in latent space between $A$#$\pi_x$ and $\pi_{y^\delta}$ to ensure fidelity to the measurements. On the contrary, we penalize the reconstruction error of the sought end-to-end reconstruction $G_\phi$. As a consequence, in our model, the measurement fidelity and the data regularization provided by $W_1((G_\phi)$#$\pi_{y^\delta}, \pi_x)$ are coupled. This allows a straightforward optimization procedure that does not involve the computation of an error bound, c.f. Proposition 3.1 in Sim, Byeongsu, et al., allows for theoretical stability results
>     and provides usable dual variables $R_{\theta^*}$ during adversarial training. Moreover, the error bound that is considered by Sim, Byeongsu, et al. in eq. (3.8) is a typical CycleGAN penalty that forces the end-to-end reconstruction $G_\phi$ to be close to the inverse of $A$. On the contrary, in UAR we have used the reconstruction error in data space. As in inverse problems the measurement operator $A$ is expected to be non-invertible, we believe that a CycleGAN penalty is too constraining for learning the end-to-end reconstruction $G_\phi$. The reconstruction error employed in UAR allows $G_\phi$ to be any right inverse of the measurement operator $A$, facilitating a better reconstruction and allowing for a more straightforward theoretical analysis.
> Besides such considerations, we also believe that our paper is the first instance of combining general end-to-end adversarial reconstructions $G_{\phi^*}$ with the learning of a variational regularizer $R_{\theta^*}$, studying this interplay from a numerical and theoretical point of view. Note that the optimal $R_{\theta^*}$ that is learnt simultaneously with the end-to-end reconstruction $G_{\phi^*}$ is used as a regularizer in the minimization problem
> \begin{align}
> \qquad \qquad \qquad  \qquad \qquad \qquad \qquad \qquad \min_x ||A(x) - y^\delta ||^2 + \lambda R_{\theta^*}(x)
> \end{align}
> to improve the end-to-end reconstruction provided by $G_{\phi^*}(y^\delta)$ and set its solution onto a robust theoretical footing.
> In particular, $R_{\theta^*}$ has the property that a gradient descent step taken in the direction $\nabla R_{\theta^*}$ pushes the vanilla reconstruction $G_{\phi^*}(y^\delta)$ closer to the data manifold. Numerical results suggest that the incorporation of this regularization step in the reconstruction considerably improves over the end-to-end reconstruction.
> Regarding the other reference (TDV) pointed out by the reviewer that has been already cited in our manuscript (reference [12]), we believe that the techniques to learn the regularizer for a specific inverse problem are fairly different than ours, not dealing with optimal transport analysis, but with learning the dynamics of a given gradient flow by formulating an optimal control problem.
>
>
> &NewLine;
> &NewLine;
>
> 2. Although our method is applicable to any inverse problem, we chose to focus on sparse-view CT since it is a prototypical inverse problem for which there are not many unpaired training approaches available. Our goal was to demonstrate the potential of our method, showing that it is capable of achieving state-of-art performances on this specific task. However, we understand the referee's concerns regarding the applicability of UAR to different inverse problems. To address this, we have conducted new experiments to demonstrate the applicability of UAR to two important inverse problems, namely image inpainting (on MNIST) and denoising (on STL-10). The results of these experiments are reported on the anonymous github repository https://github.com/uar-new-results/uar_new_results. We emphasize that these experiments serve as proof-of-concept and one can make UAR competitive with the state-of-the-art algorithms for a specific inverse problem by carefully fine-tuning the hyper-parameters associated with training and the network architectures.

---

> ### Comment · Reviewer_QSGm · 2021-08-13
> **Post Rebuttal**
>
> The authors have made significant effort trying to address the points raised. I would like to increase my score to 6.

---

### Decision · Program_Chairs · 2021-09-27

**Decision:**

Accept (Poster)

**Comment:**

A solid paper proposing a new method combining end-to-end supervised training with learned regularization in a variational setting. The method adversarially trains the weights of a deep neural network (used as a prior for regularization) while also learning an end-to-end, unrolled deep network.

Reviewers liked the manuscript and were satisfied by the long rebuttals and detailed conversation.